# STOCHASTIC SELF-GUIDANCE FOR TRAINING-FREE ENHANCEMENT OF DIFFUSION MODELS

**Chubin Chen[1,*]    Jiashu Zhu[2,*]    Xiaokun Feng[3]    Nisha Huang[1]    Chen Zhu[2]**
**Meiqi Wu[3]    Fangyuan Mao[3]    Jiahong Wu[2,‡]    Xiangxiang Chu[2]    Xiu Li[1,†]**

[1]Tsinghua University    [2]AMAP, Alibaba Group    [3]CASIA

*Project Page*: *https://s2guidance.github.io/*

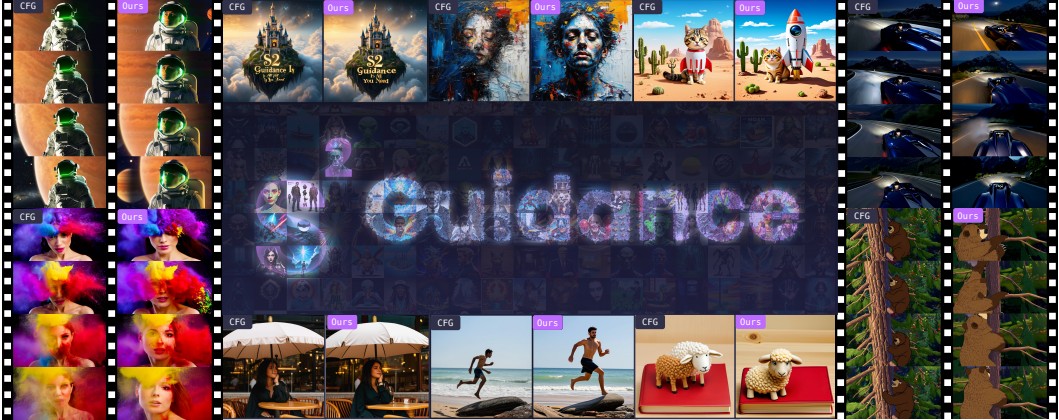

Figure 1: **Visual results of $S^2$-Guidance versus CFG.** Our proposed method $S^2$-Guidance significantly elevates the quality and coherence of both T2I and T2V generation. **Observe (in examples surrounding the center):** Our method produces generations with **superior temporal dynamics**, including more pronounced motion (bear) and dynamic camera angles that convey speed (car). It renders **finer details**, such as the astronaut's transparent helmet and rich facial details, and creates images with **fewer artifacts** (runner, woman with umbrella), **richer artistic detail** (abstract portrait, castle, colored powder exploding), and **improved object coherence** (cat and rocket, sheep). See Appendix B.5 for our prompts.

## ABSTRACT

Classifier-free Guidance (CFG) is a widely used technique for improving conditional generation in diffusion models. However, our empirical analysis of both Gaussian mixture data and real-world image data distributions reveals a discrepancy between the suboptimal results produced by CFG and the ground truth. The model's excessive reliance on these suboptimal predictions often leads to low fidelity and semantic incoherence. To address this issue, we first empirically demonstrate that the model's suboptimal predictions can be effectively rectified using sub-networks of the model itself, without requiring additional training or the integration of external modules. Building on this insight, we propose $S^2$-Guidance (*S*tochastic *S*elf-Guidance), a novel method that leverages stochastic block-dropping during the denoising process to activate sub-networks for self-guidance. This approach effectively steers the sampling trajectory towards high-quality regions. Comprehensive experiments, including on class-conditional ImageNet generation and across multiple benchmarks for text-to-image and text-to-video generation, demonstrate the superiority of $S^2$-Guidance. Both qualitative and quantitative results show that $S^2$-Guidance consistently surpasses CFG and other advanced guidance strategies. Our code will be released.

---

*Equal contribution.
†Corresponding author.
‡Project lead.

# 1 INTRODUCTION

Diffusion models (Song et al., 2020a; Ho et al., 2020) have enabled rapid advances in high-quality text-to-image (Rombach et al., 2022; Podell et al., 2023; Labs, 2024; Wu et al., 2025a) and text-to-video (Wan et al., 2025; Kong et al., 2024) generation. A key driver of this success is the advent of conditional guidance techniques, which steer the generation process to enhance adherence to given conditions. However, naively applying the conditioning signal often proves insufficient (Dhariwal & Nichol, 2021). Classifier-free Guidance (CFG) (Ho & Salimans, 2022) has become the mainstream approach for improving conditional generation. It employs a Bayesian implicit classifier to prioritize conditional probability, enhancing adherence to conditions and image quality. However, despite its effectiveness, it often results in semantic incoherence and a loss of fine details, as shown in Figure 1.

Recent studies (Chung et al., 2024; Sadat et al., 2024; Fan et al., 2025; Kynkäänniemi et al., 2024; Jin et al., 2025) have further explored methods to improve guidance. Although these methods improve quality to some extent, they primarily address specific issues while leaving the underlying mechanisms of CFG unexplored. A representative work that begins to explore this issue is Autoguidance (Karras et al., 2024), which identifies deficiencies in the model's training objective and proposes using a weak model for guidance. Subsequent works (Hong et al., 2023; Ahn et al., 2024; Jeon, 2025; Hong, 2024) propose modifying specific attention regions to mimic a weak model for various tasks (Qi et al., 2023; Simsar et al., 2024). However, these methods either require training to acquire the weak model or rely on empirical, task-specific modifications to the network, which in turn demand meticulous hyperparameter tuning.

To address this, we first analyze the suboptimal results produced by CFG and the underlying mechanisms of weak-model guidance. Specifically, our analysis begins with a toy example on Gaussian mixture modeling, where a closed-form solution allows for precise evaluation against the ground truth (Brown et al., 2022; Pope et al., 2021), and is subsequently validated on real-world image data. Furthermore, we observe that applying stochastic block-dropping during the model's forward process produces results highly similar to the weak model used in Autoguidance. Building on this discovery, we propose $S^2$-Guidance, a simple yet effective approach to address the suboptimal predictions of CFG and guide sampling towards higher quality and fidelity. Unlike prior methods that rely on externally trained or manually tuned weak models, $S^2$-Guidance leverages the model's own intrinsic structure in a training-free manner, effectively steering the denoising trajectory away from failure modes to enhance the performance of conditional diffusion models.

Our contributions are summarized as follows:

**(i)** We first analyze the guidance behavior of CFG and the underlying mechanisms of weak-model guidance through a series of toy examples. These examples allow us to visually analyze the suboptimal results of CFG. Empirical observations reveal that the sampling trajectory can be effectively rectified by the model's own sub-networks, which exhibit guidance behavior similar to that of a weak model.

**(ii)** We propose $S^2$-Guidance, a novel method that leverages stochastic block-dropping during the forward process to activate sub-networks for self-guidance, thereby bypassing the need to construct weak models through additional training or a trial-and-error manual selection process. Furthermore, we demonstrate that in the iterative denoising process, a single block-dropping per timestep is sufficient to steer the sampling trajectory towards high-quality regions. This approach achieves strong performance while substantially reducing computational costs compared to the naive variant.

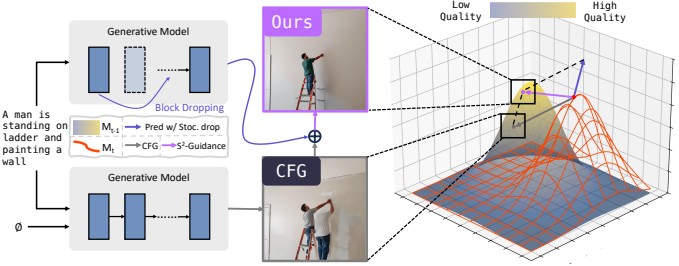

Figure 2: **An illustration of our guidance mechanism on the generation quality manifold.** Unlike suboptimal CFG guidance (gray), $S^2$-Guidance derives a corrective signal (blue) via stochastic block-dropping, steering the generation update (purple) toward the optimal quality peak (yellow).

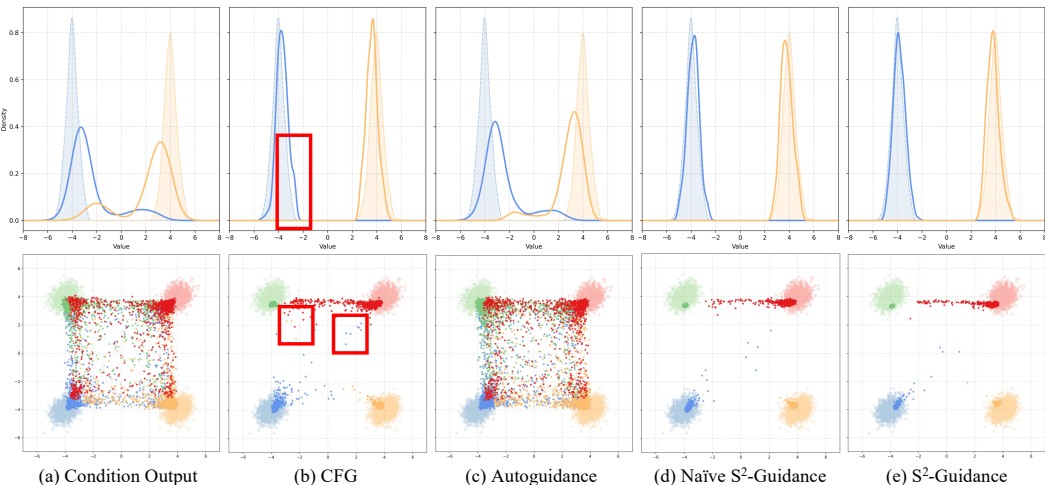

Figure 3: $S^2$-**Guidance successfully balances guidance strength and distribution fidelity.** Comparison on 1D (top) and 2D (bottom) toy examples. Unlike CFG, which distorts the sample distribution (see red boxes), or other methods that fail to separate modes, $S^2$-Guidance accurately captures both the location and shape of the ground truth distributions (semi-transparent).

**(iii)** Our method can be seamlessly adapted to various diffusion models. Comprehensive experiments—on class-conditional ImageNet generation and across multiple benchmarks for text-to-image and text-to-video tasks—establish the superiority of $S^2$-Guidance. Both qualitative and quantitative results confirm that $S^2$-Guidance consistently surpasses not only CFG but also other advanced guidance strategies.

## 2 BACKGROUND

**Diffusion Models.** Diffusion models (Croitoru et al., 2023; Peebles & Xie, 2023; Esser et al., 2024; Chu et al., 2025b) are a class of powerful generative models that learn to reverse a predefined forward process, which gradually perturbs data $x_0$ into Gaussian noise $x_T$. The reverse process is typically governed by a time-reversed stochastic differential equation (SDE) (Song et al., 2020b), which relies on accurately estimating a score function $\nabla_{x_t} \log p_t(x_t)$ using a neural network $D_\theta$. Flow-based models (Lipman et al., 2023; Liu et al., 2022; Gat et al., 2024) can also be viewed as a special class of diffusion models, as they both aim to learn a continuous transformation between a simple prior distribution and the complex data distribution (Gao & Zhu, 2025). In practical applications (Huang et al., 2023a; Zhu et al., 2024; Huang et al., 2024a; Mao et al., 2025), generation is often conditioned on signals $c$ (*e.g.*, text prompts), shifting the objective to modeling the conditional score $\nabla_{x_t} \log p_t(x_t|c)$.

**Classifier-free Guidance (CFG).** CFG (Ho & Salimans, 2022) has become the cornerstone for controllable generation (Huang et al., 2023b; Wang et al., 2024b; Fang et al., 2025; He et al., 2025; Ma et al., 2024b; 2025a) by offering a simple yet effective mechanism to enhance conditioning. Instead of only using the conditional prediction $D_\theta(x_t|c)$, CFG forms a guided score by extrapolating from an unconditional one $D_\theta(x_t|\phi)$:

$$\tilde{D}_\theta^\lambda(x_t|c) = D_\theta(x_t|\phi) + \lambda \left( D_\theta(x_t|c) - D_\theta(x_t|\phi) \right), \tag{1}$$

where $\lambda$ is the guidance scale. However, despite its effectiveness, this approach suffers from notable drawbacks (Sadat et al., 2024; Hong et al., 2023; Karras et al., 2024), including semantic inconsistencies and a significant loss of fine-grained details, as illustrated in Figure 1.

**Weak-model Guidance.** A promising direction to improve CFG is to leverage an auxiliary "weak" model to refine the guidance signal. For instance, Autoguidance (Karras et al., 2024) employs a separately trained, degraded version of the full model, but such models are often infeasible to obtain for large-scale pretrained models. To circumvent this, recent works simulate a weak model

|  |  |  |  |  |  |
|---|---|---|---|---|---|
| (a) Condition Output | (b) CFG | (c) Autoguidance | (d) Naïve S²-Guidance | (e) S²-Guidance | (f) Visualization |

Figure 4: $S^2$-**Guidance avoids the distributional collapse of CFG on CIFAR-10.** t-SNE shows generated features (points) vs. real data (contours). CFG (b) exhibits severe collapse, whereas $S^2$-Guidance (e) preserves the distribution's structure while ensuring class separation. See (f) for qualitative examples.

by modifying the model's architecture or perturbing its internal states. For instance, some studies rely on heuristic perturbations like attention-guided blurring of predicted samples (Hong et al., 2023; Ahn et al., 2024); SEG (Hong, 2024) later proposes an alternative from an energy-based perspective; and other works develop strategies for specific tasks (Jeon, 2025; Hyung et al., 2025) However, these perturbation techniques often rely on task-specific, hand-crafted architectural modifications, which limits their generalizability. In contrast, as shown in Figure 2, our $S^2$-Guidance introduces a novel and flexible approach. We guide the sampling process by dynamically activating sub-networks via stochastic block dropping, thereby avoiding the need to construct a weak model through auxiliary training or manually designed perturbation schemes.

## 3 METHODOLOGY

### 3.1 VISUALIZING AND REVISITING WEAK-MODEL GUIDANCE

We begin by visualizing the suboptimal outcomes of CFG using Gaussian mixture data (Ho & Salimans, 2022), a toy example with closed-form solutions. This allows us to systematically observe the discrepancies between predictions and ground truth. Building on the analysis of how weak-model guidance (Karras et al., 2024) improves results, we identify its limitations and propose incorporating stochastic sub-networks into the CFG framework, providing a novel approach to enhance model performance.

CFG improves conditional generation by implicitly amplifying the conditional probability density, raising it to a power greater than one (Bradley & Nakkiran, 2024). Figure 3 illustrates a 1D toy example (top) aimed at learning a Gaussian Mixture distribution with modes at $-4$ and $4$. While CFG significantly improves the baseline conditional output, it also introduces a notable drawback: as highlighted by the red box, the mode of the generated distribution is slightly shifted from the ground truth. A similar shift occurs in a 2D toy example (bottom), where samples are scattered into unintended regions. These findings suggest that, although CFG enhances sample quality, its distributional fidelity remains suboptimal. Autoguidance, as a representative of weak-model guidance (Karras et al., 2024; Hong et al., 2023; Hong, 2024; Ahn et al., 2024), is designed to guide the model toward well-learned, high-probability regions by leveraging a weak model. As shown in Figure 3 (middle), AutoGuidance improves the peak near -4 but remains limited. Its improvement stems from the construction of a weak model, with the extent of enhancement depending on the weak model's effectiveness. Such models are typically created by reducing model capacity or training epochs.

However, this approach faces practical limitations that restrict its broader applicability. First, relying on externally designed weak models poses scalability challenges, as obtaining a reduced version trained for fewer epochs alongside a large-scale pretrained model is often impractical. Second, as highlighted by (Karras et al., 2024), selecting an appropriate weak model is constrained by various factors. Once chosen, the weak model affects the entire denoising process, limiting the flexibility of guidance. A poorly designed weak model fails to effectively prevent low-quality outputs (Hong, 2024), as shown in Figure 3 (c), where guided outputs still deviate notably from the target distribution.

This raises an important question: *Can we eliminate the reliance on externally prescribed weak models while still identifying error-prone regions?* Prior works (Lou et al., 2024; Avrahami et al., 2025;

Yuan et al., 2024) have shown that mainstream generative architectures, such as DiT (Peebles & Xie, 2023; Chu et al., 2024), exhibit significant redundancy, as outputs across different transformer blocks often show high similarity (Chen et al., 2024). Inspired by this, we hypothesize that sub-networks within such architectures can function as weak models, capturing outputs similar to the full model but with more pronounced errors. By leveraging these sub-network predictions, we aim to refine existing CFG, effectively steering the model away from suboptimal outputs. The following subsections present a detailed description of our approach along with its empirical validation.

## 3.2 NAIVE $S^2$-GUIDANCE

Building on the preceding observation, our key insight is that *we can leverage the model's own sub-networks to intrinsically steer the denoising trajectory away from potential failure modes, thereby refining the suboptimal results of CFG.*

As revealed in Autoguidance (Karras et al., 2024), problems in generative models depend on various factors (e.g., network architecture, dataset properties, etc.), making it difficult to pinpoint which components play a decisive role. Therefore, it is challenging to *a priori* define an optimal sub-network that best captures low-quality regions. Motivated by (Gal & Ghahramani, 2016), a naive solution is to leverage as many diverse stochastic sub-networks as possible to construct multiple weak models. These weak models then guide the main model away from low-quality regions during each forward pass by steering it away from their outputs. We refer to this approach as Naive *S*tochastic *S*ub-network Guidance (Naive $S^2$-Guidance). Intuitively, this can be understood as applying stochastic "dropout" to different blocks, constructing various sub-networks that capture diverse low-probability regions.

Specifically, for a given binary mask $\mathbf{m}$, sampled via stochastic block-dropping from the induced distribution $p(\mathbf{m})$, the weak model's prediction is defined as:

$$\hat{D}_\theta(x_t \mid c, \mathbf{m}) = D_\theta(x_t \mid c; \boldsymbol{\theta} \odot \mathbf{m}), \tag{2}$$

where $\mathbf{m}$ determines which blocks of the network parameters $\boldsymbol{\theta}$ are activated, forming a latent sub-network during each forward pass. Naive $S^2$-Guidance is then expressed as:

$$\tilde{D}_\theta^\lambda(x_t \mid c) = D_\theta(x_t \mid \phi) + \lambda\big(D_\theta(x_t \mid c) - D_\theta(x_t \mid \phi)\big)$$
$$- \frac{\omega}{N} \sum_{i=1}^N \big(\hat{D}_\theta(x_t \mid c, \mathbf{m}_i) - D_\theta(x_t \mid c)\big), \tag{3}$$

where $\mathbf{m}_i \sim p(\mathbf{m})$ is the binary mask for the $i$-th stochastic sub-network, $\omega$ controls the strength of the self-guidance, referred to as the $S^2$ Scale. $\hat{D}_\theta(x_t \mid c, \mathbf{m}_i)$ represents the prediction from the $i$-th sampled sub-network, and the self-guidance signal is defined as its deviation from the full-model prediction. $N$ denotes the total number of latent sub-networks sampled during each forward pass.

For the sampling distribution $p(\mathbf{m})$, a crucial consideration is to ensure its effectiveness and generalizability across different models. Our approach is predicated on the principle of identifying and preserving the model's structurally critical components. Based on empirical analysis, we exclude these key blocks from the dropping process and then sample a proportion of the remaining blocks to be dropped. Detailed ablations regarding the stochastic sampling design and its stability are provided in Section 4.

To validate our hypothesis, we conduct experiments on toy examples with 1D and 2D Gaussian mixture data, as well as on real-world datasets (see Appendix B.1.1 for more details). As shown in Figure 3 (d), compared to the original CFG, our Naive $S^2$-Guidance not only leads to predictions that better fit the target distribution but also mitigates the drift phenomenon, thereby improving fidelity. This demonstrates that our method effectively refines the suboptimal results of CFG. Furthermore, compared to Autoguidance, $S^2$-Guidance eliminates the need for explicitly constructing weak models. By adopting this simple yet effective approach, it avoids generating results that lie in intermediate regions, thereby reducing mode confusion. These results provide strong empirical evidence that leveraging Naive $S^2$-Guidance can significantly enhance both the quality and robustness of conditional generation.

| Model | Method | HPSv2.1 (%) ↑ | | | | | T2I-CompBench (%) ↑ | | | Qalign ↑ | |
|---|---|---|---|---|---|---|---|---|---|---|---|
| | | Anime | Concept | Paint. | Photo | Avg. | Color | Shape | Texture | HPSv2.1 | T2I-Comp. |
| SD3 | CFG | 31.55 | 30.87 | 31.22 | 28.27 | 30.48 | 53.61 | 51.20 | 52.45 | 4.66 | 4.74 |
| | CFG++ | 31.57 | 30.76 | 30.96 | 27.54 | 30.21 | 46.39 | 47.18 | 46.33 | **4.68** | 4.73 |
| | APG | 30.77 | 30.18 | 30.53 | 27.12 | 29.65 | 45.28 | 46.27 | 46.84 | **4.68** | 4.73 |
| | CFG-Zero | 31.99 | 31.17 | 31.42 | 28.54 | 30.78 | 52.70 | 52.84 | 53.37 | 4.66 | **4.77** |
| | SEG | 31.20 | 30.56 | 31.07 | 28.74 | 30.39 | 58.20 | 57.68 | **57.17** | 4.33 | 4.45 |
| | **Ours** | **32.14** | **31.32** | **31.70** | **29.19** | **31.09** | **59.63** | **58.71** | 56.77 | 4.65 | 4.74 |
| SD3.5 | CFG | 32.34 | 31.51 | 31.50 | 27.93 | 30.82 | 51.29 | 47.71 | 47.39 | 4.63 | 4.66 |
| | CFG++ | 31.99 | 31.02 | 31.36 | 27.32 | 30.42 | 38.05 | 37.52 | 34.87 | 4.65 | 4.58 |
| | APG | 31.43 | 30.74 | 31.12 | 27.07 | 30.09 | 35.67 | 37.86 | 35.67 | 4.68 | 4.65 |
| | CFG-Zero | 32.77 | 31.91 | 31.95 | 28.27 | 31.23 | 52.01 | 46.99 | 48.36 | 4.66 | 4.70 |
| | SEG | 31.77 | 31.30 | 31.40 | 28.34 | 30.71 | **57.59** | **55.52** | **54.03** | 4.41 | 4.45 |
| | **Ours** | **32.89** | **32.15** | **32.28** | **28.94** | **31.56** | 57.57 | 51.23 | 50.13 | **4.70** | **4.74** |

Table 1: **Quantitative comparison in T2I generation.** Our method establishes a new state-of-the-art, demonstrating significant improvements even on highly competitive benchmarks. On **HPSv2.1**, a benchmark where score margins are typically narrow, $S^2$-Guidance consistently outperforms all baselines across every individual dimension. This lead is even more pronounced on **T2I-CompBench**, where our approach shows substantial gains in compositional attributes like Color and Shape. Notably, $S^2$-Guidance also achieves the highest or near-highest aesthetic scores (**Qalign**) on both benchmarks, demonstrating its superior performance in visual quality. Higher scores (↑) are better. Best results are in **bold**; second-best are underlined.

## 3.3 $S^2$-Guidance Is Sufficient

However, Naive $S^2$-Guidance incurs significant computational overhead, which severely limits its practicality. In the process of constructing sub-networks, we find that constraining stochastic block-dropping within a specific range allows sub-networks, even those generated by dropping at different blocks, to consistently guide the model toward the ideal distribution (Figure 9).

Therefore, we propose a simplified approach: performing a single stochastic block-dropping operation at each timestep for self-guidance. We refer to this approach as $S^2$-Guidance, which achieves highly competitive results. At timestep $t$, $S^2$-Guidance is expressed as:

$$\tilde{D}_\theta^\lambda(x_t \mid c) = D_\theta(x_t \mid \phi) + \lambda\big(D_\theta(x_t \mid c) - D_\theta(x_t \mid \phi)\big)$$
$$- \omega\big(\hat{D}_\theta(x_t \mid c, \mathbf{m}_t) - D_\theta(x_t \mid c)\big). \tag{4}$$

The overall algorithm is summarized in Algorithm 1.

We empirically validate the proposed $S^2$-Guidance on toy examples with 1D and 2D Gaussian mixture data, as well as on real-world datasets. As shown in Figure 3 (e), $S^2$-Guidance performs comparably to Naive $S^2$-Guidance. On both 1D and 2D Gaussian mixture distributions, it produces results that closely align with the ideal distribution, while exhibiting efficiency without significant degradation. Moreover, as illustrated in Figure 4 (e, f), $S^2$-Guidance achieves highly competitive performance on

---

**Algorithm 1** $S^2$-Guidance

**Require:** Trained denoiser $D_\theta$, initial noise $x_T$, guidance scale $\lambda$, $S^2$ scale $\omega$, number of timesteps $T$.
1: **for** $t = T, \dots, 1$ **do**
2:      $m_t \leftarrow$ GenerateStochasticMask() # Generate mask
3:      $D_{\text{uncond}} \leftarrow D_\theta(x_t, \phi, t)$
4:      $D_{\text{cond}} \leftarrow D_\theta(x_t, c, t)$
5:      $\hat{D}_s \leftarrow D_\theta(x_t, c, t, m_t)$ # Sub-network prediction
6:      $\tilde{D} \leftarrow D_{\text{uncond}} + \lambda(D_{\text{cond}} - D_{\text{uncond}}) - \omega(\hat{D}_s - D_{\text{cond}})$
7:      $x_{t-1} \leftarrow$ SchedulerStep($\tilde{D}, x_t, t$)
8: **end for**
9: **return** $x_0$

---

real-world datasets, highlighting its practical effectiveness. To further analyze the stochastic block-dropping strategy, we conduct a detailed experimental study in Section 4.5. Our empirical analysis reveals that, when the block drop ratio is maintained around 10% of the network's blocks, the resulting sub-networks consistently enable the model to achieve better performance. This strategy proves effective across mainstream DiT (Peebles & Xie, 2023) architectures, leveraging the redundancy in the outputs to dynamically construct diverse stochastic sub-networks. Unlike explicitly constructed

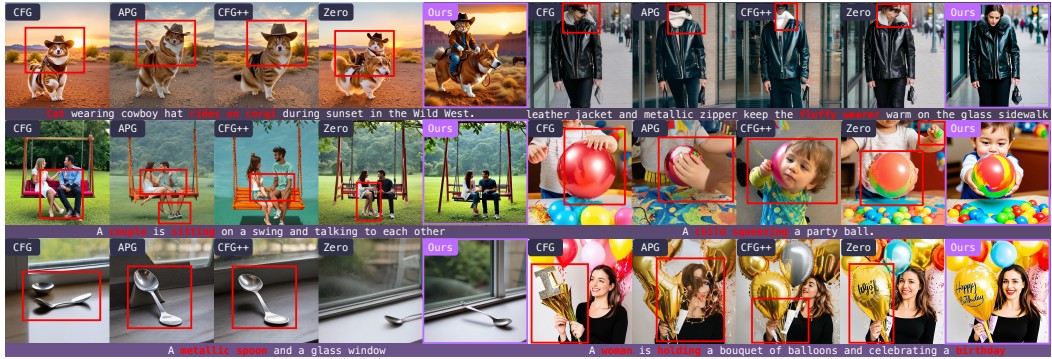

Figure 5: $S^2$-**Guidance consistently generates superior images in both aesthetic quality and prompt coherence.** While existing guidance methods often produce artifacts, distorted objects, or fail to follow complex prompts (see red boxes), our approach yields clean, coherent, and visually pleasing results without such flaws.

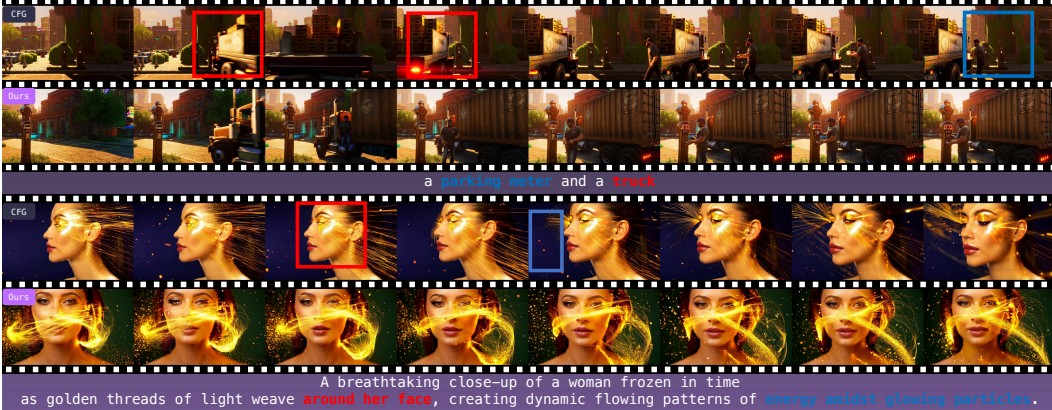

Figure 6: $S^2$-**Guidance generates temporally coherent and physically plausible videos, overcoming key failures of CFG. Top Row:** CFG struggles with plausible motion, depicting a truck that unnaturally slides sideways instead of driving forward (red boxes). Our method renders a stable and realistic scene. **Bottom Row:** CFG fails to capture the full prompt, as the light does not weave "around her face" (red box) and lacks "glowing particles" (blue box). $S^2$-Guidance faithfully produces a dynamic, visually rich scene adhering to the complex description.

weak models, which once selected affect the entire denoising process, stochastic block-dropping enables the creation of sub-networks independently at different timesteps. This dynamic diversity introduces self-guidance throughout the diffusion process, allowing predictions to evolve iteratively and steering the outputs toward higher-quality results.

## 4 EXPERIMENTS

### 4.1 IMPLEMENTATION DETAILS

**Benchmark.** We perform comprehensive evaluations across three tasks: class-conditional image generation, text-to-image (T2I) and text-to-video (T2V) generation. For class-conditional generation, we use ImageNet at a $256 \times 256$ resolution. For T2I evaluation, we use two popular benchmarks: HPSv2.1 (Wu et al., 2023b), a benchmark designed to evaluate alignment with human preferences across 3,200 prompts in four styles, and T2I-CompBench (Huang et al., 2023a) for assessing performance in complex scenes. In addition to the benchmark-specific evaluation metrics, we employ Qalign (Wu et al., 2023a) to compute aesthetic scores for a more comprehensive assessment. For T2V evaluation (Liu et al., 2024; Ling et al., 2025; Feng et al., 2025a; Chen et al., 2025c), we adopt the standard prompts and evaluation metrics provided by VBench (Huang et al., 2024b).

| Model | Method | Total Score | Quality Score | Semantic Score | Subject Consistency | Background Consistency | Aesthetic Quality | Imaging Quality | Object Class | Appearance Style |
|---|---|---|---|---|---|---|---|---|---|---|
| Wan1.3B | CFG | 80.29 | 84.32 | 64.16 | 96.53 | 95.46 | **60.52** | 67.65 | 77.06 | 20.15 |
| | CFG++ | 80.35 | 83.58 | **67.43** | **96.70** | 93.28 | 59.02 | **69.14** | 70.06 | 19.75 |
| | APG | 70.83 | 77.13 | 45.61 | 96.45 | 95.39 | 49.42 | 64.39 | 59.02 | 20.01 |
| | STG | 78.78 | 83.92 | 58.19 | 95.03 | **96.04** | 59.03 | 65.59 | 68.20 | **21.51** |
| | CFG-Zero | 80.71 | 84.51 | 65.53 | 96.33 | 94.56 | 59.69 | 69.05 | **78.16** | 20.31 |
| | **Ours** | **80.93** | **84.74** | 65.70 | 96.57 | 95.80 | 60.52 | 68.19 | 78.09 | 20.59 |
| Wan14B | CFG | 82.65 | 84.88 | 73.76 | **94.45** | **97.66** | 68.68 | **67.82** | 84.97 | 22.14 |
| | **Ours** | **82.84** | **84.89** | **74.65** | 94.21 | 97.56 | **68.78** | 67.77 | **89.08** | **22.27** |

Table 3: **Quantitative comparison on VBench.** $S^2$-Guidance consistently outperforms mainstream methods on both Wan-1.3B and Wan-14B models. While evaluated on all 16 dimensions, this table shows a representative subset of 9 key metrics. Our method achieves the highest **Total Score** and demonstrates significant improvements. Best results are in **bold**; second-best are underlined.

**Baselines.** For T2I task, we employ the high-performing Stable Diffusion 3 (SD3) (Esser et al., 2024) and SD3.5 (AI, 2024). For T2V task, we utilize the latest Wan-1.3B and Wan-14B models (Wan et al., 2025). Furthermore, to demonstrate the versatility of our guidance approach, we conduct a comparative analysis not only against original CFG but also with five state-of-the-art methods: CFG++ (Chung et al., 2024), CFG-Zero (Fan et al., 2025), APG (Sadat et al., 2024), STG (Hyung et al., 2025) and SEG (Hong, 2024). See Appendix B.2 for additional evaluations and Appendix B.4 for implementation details.

## 4.2 CLASS-CONDITIONAL IMAGENET GENERATION

Evaluated on ImageNet $256 \times 256$ with a pretrained SiT-XL model (Ma et al., 2024a), $S^2$-Guidance demonstrates clear superiority over both CFG and other advanced guidance strategies (many of which are not designed for advanced flow-based models and thus struggle to perform well (Fan et al., 2025)). As shown in Table 2, our method achieves the best performance, attaining both the highest Inception Score of **259.22** for image diversity and fidelity, and the lowest FID of **2.08** for perceptual quality and distributional alignment.

| Method | IS↑ | FID↓ |
|---|---|---|
| Baseline | 125.13 | 9.41 |
| w/ CFG | 258.09 | 2.15 |
| w/ ADG | 257.92 | 2.37 |
| w/ CFG++ | 257.04 | 2.25 |
| w/ SEG | 258.35 | 2.29 |
| w/ CFG-Zero | 258.87 | 2.10 |
| **w/ Ours** | **259.12** | **2.03** |

Table 2: **Quantitative evaluation on ImageNet** $256 \times 256$ **dataset**.

## 4.3 TEXT-TO-IMAGE GENERATION

The quantitative comparisons are presented in Table 1. On HPSv2.1, $S^2$-Guidance achieves the best performance not only in average scores but also across all individual dimensions, demonstrating the effectiveness of our method. By steering the sampling trajectory away from suboptimal paths inherent in CFG, $S^2$-Guidance achieves better alignment with human preferences. The performance on T2I-CompBench further highlights the strength of our approach, showcasing its effectiveness in handling complex generation tasks. Moreover, the high aesthetic scores confirm our method's ability to produce images with superior visual appeal.

The qualitative comparisons are presented in Figure 5. Compared to CFG and other methods, $S^2$-Guidance achieves significant improvements in both visual quality and semantic coherence: it produces higher-quality images with finer details and better semantic alignment with text descriptions, a result consistent with our toy examples.

## 4.4 TEXT-TO-VIDEO GENERATION

The quantitative comparisons are presented in Table 3. On the Wan-1.3B model, $S^2$-Guidance achieves the highest **Total Score** (80.93), outperforming all baselines. We further conduct experiments on the larger Wan-14B model, demonstrating significant improvements compared to CFG.

The quantitative comparisons are presented in Figure 6. Our method generates videos with substantially improved quality and coherence. The examples highlight that $S^2$-Guidance effectively addresses two critical failures of original CFG: the loss of physical plausibility in object motion and the inability to adhere to complex, compositional prompts. Consequently, our approach yields

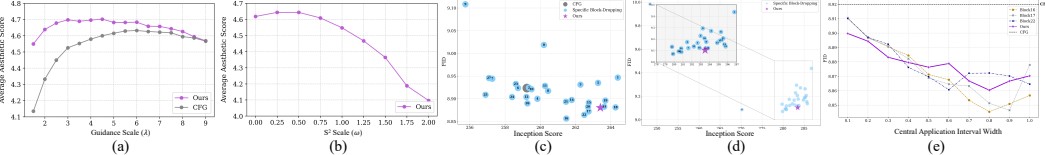

Figure 7: **Comprehensive ablation analysis of $S^2$-Guidance.** **(a)** Comparing aesthetic scores of $S^2$-Guidance and CFG across various guidance scales ($\lambda$). **(b)** Analyzing aesthetic scores of $S^2$-Guidance across various self-guidance scales ($\omega$). **(c, d)** Comparison of our stochastic block-dropping strategy against dropping a single, fixed block for the SiT and DiT architectures, respectively. Performance is measured by the FID-IS trade-off, where the lower-right corner indicates a better balance. **(e)** Ablation on the application range of $S^2$-Guidance. The x-axis represents the width of the central interval of of noise levels where block-dropping is applied (e.g., a value of 0.2 corresponds to the central 20% of the denoising process).

videos that are not only more physically realistic but also demonstrate superior prompt coherence, faithfully realizing the user's creative intent.

We further perform user study for both T2I and T2V generation. Our method is significantly preferred over all baselines in terms of both visual quality and prompt alignment. Full details are presented in Appendix B.3.

## 4.5 ABLATION STUDY

**Performance across Different Guidance Scales.** We conduct experiments to compare $S^2$-Guidance with CFG across various guidance scales, focusing on aesthetic scores. As shown in Figure 7 (a), $S^2$-Guidance consistently outperforms CFG across a wide range of scales. Unlike CFG, which shows significant performance variance depending on the scale, our method exhibits stability and achieves high performance with minimal sensitivity to guidance scales. Notably, in most cases, our method even surpasses the best performance achieved by CFG, demonstrating its robustness.

**Analysis of Block-dropping Strategy.** We conduct a series of experiments to thoroughly analyze the effectiveness and robustness of our block-dropping strategy. First, to investigate the **importance of individual block**, we perform experiments on diverse model architectures. We drop a single, specific block throughout the entire denoising process to obtain the sub-network prediction and compare its performance against our method. As shown in Figure 7 (c,d), dropping the initial block consistently leads to performance degradation across both models. However, for the remaining blocks, this block-wise ablation does not yield a universal rule. We find that the optimal block to drop varies significantly across different architectures, a challenge also highlighted by AutoGuidance (Karras et al., 2024). In contrast, our method eliminates the need for such complex tuning. Its simple stochastic strategy automatically outperforms most meticulously selected fixed configurations. Furthermore, inspired by (Kynkäänniemi et al., 2024), we analyze the **optimal application interval** for block-dropping. As shown in the Figure 7 (e), applying block-dropping within the central 80% interval of noise levels yields robust performance. Our method reduces FID compared to CFG and often outperforms the top-performing configurations derived from prior block-wise ablation.

**Effect of $S^2$ Scale $\omega$.** We conduct experiments to analyze the scale of $S^2$-Guidance $\omega$ , as shown in Figure 7 (b). When $\omega$ is set to a smaller value, it improves the aesthetic score. However, since CFG has already produced a suboptimal result, using a larger $\omega$ tends to overadjust, leading to a decline in quality.

**Analysis of $S^2$-Guidance and Naive $S^2$-Guidance.** We compare applying block-dropping once versus multiple times per sampling step. Empirically, increasing the number of applications yields diminishing returns in aesthetic scores (see Figure 8 in Appendix). We therefore conclude that a single application per timestep is sufficient, striking an effective balance between high performance and computational efficiency. See Appendix A.1 for the theoretical analysis and Appendix B.1.2 for further visualizations.

**Effect of Drop-Ratio.** We investigate the impact of the drop-ratio on the SD3.5 model with 24 blocks. As shown in Table 4, when the number of dropped blocks is limited to 3/24 (approximately 10%), the aesthetic score remains stable at a relatively high level. However, dropping more blocks leads to a gradual decline in performance. Empirically, we observe that a drop-ratio of about 10% significantly improves performance.

**Visual Analysis of Block-Dropping Impact** To intuitively address concerns about how dropping blocks affects the final output, we provide a visual analysis in Figure 15 (Appendix B.2). This figure presents the results of an extreme test case on the SiT-XL model for the ImageNet 256×256 task. In this setup, for each of the 28 generated images, a single, fixed transformer block was dropped for the entire duration of the inference process. As can be observed, the resulting images exhibit remarkable visual consistency and coherence, with no single dropped block leading to severe artifacts or a collapse in quality. This provides compelling visual evidence of the model's inherent robustness against block-level perturbations.

| Stochastic Block-Dropping | |
|---|---|
| Num. | Aes. |
| 0 | 4.618 |
| 1 | 4.652 |
| 2 | 4.643 |
| 3 | 4.616 |
| 4 | 4.531 |

Table 4: Effect of the number of dropped blocks on aesthetic scores.

**Computational Cost and Peak Memory** We conduct a direct comparison of FLOPs, runtime, and peak memory requirements against standard CFG. The benchmark, performed on a text-to-image task with 28 inference steps, is summarized in Table 5. The results show that our $S^2$-Guidance incurs an overhead of approximately 40% in both runtime and computational cost. While this entails a notable overhead, we posit that it is justified by a superior performance-efficiency trade-off, as we demonstrate in the subsequent analysis. Notably, peak GPU memory allocation remains unchanged. This is because the two forward passes within each denoising step—one for the full model and one for the sub-network—are executed sequentially. The memory from the first pass is released before the second begins, ensuring the peak memory footprint does not exceed that of a single standard CFG evaluation.

| Method | Total Runtime (s) | Transformer FLOPs (TFLOPs) | Peak GPU Memory (GB) |
|---|---|---|---|
| CFG | 29.2 | 168.4 | ∼33.8 |
| **Ours** | 40.2 | 237.6 | ∼33.8 |

Table 5: Computational cost and memory comparison.

**Performance-Efficiency Trade-off** While our method entails a computational overhead, we argue it is justified by a superior performance-efficiency trade-off, as analyzed in Figure 14 (Appendix B.2). This figure plots the HPS Score against a normalized computational cost, where the cost for $S^2$-Guidance is scaled by a factor of 1.4 to account for its ∼40% overhead per step. The results clearly show that our method establishes a more favorable performance-efficiency frontier, consistently achieving higher performance for any given computational budget. For instance, $S^2$-Guidance with only 20 inference steps (equivalent cost of 28) surpasses the HPS score of standard CFG with 60 steps. This analysis compellingly demonstrates that our approach is a more practical and advanced choice for maximizing generation quality within a given computational budget.

## 5 CONCLUSION

In this work, we propose $S^2$-Guidance, a training-free stochastic self-guidance method that enhances diffusion transformers by improving the CFG mechanism. We first conduct an empirical analysis of CFG, revealing that it often generates suboptimal results. Building on this insight, we introduce $S^2$-Guidance, which leverages stochastic block-dropping during the forward pass to effectively guiding the model away from potential low-quality predictions, thereby improving fidelity. Theoretical analysis and extensive experiments, including class-conditional image, text-to-image and text-to-video generation across multiple models and benchmarks, demonstrate that $S^2$-Guidance delivers superior performance, consistently surpassing CFG and other advanced guidance strategies.

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

# A APPENDIX

## OVERVIEW

This appendix is divided into three main parts, covering method details, experimental supplements, and future work.

**Appendix A** The first part focuses on the details and discussions of the methodology, including:

- A principled derivation of Naive $S^2$-Guidance from a Bayesian perspective.
- A detailed analysis between $S^2$-Guidance and Naive $S^2$-Guidance.

**Appendix B** The second part provides supplementary information about experiments, covering:

- Explanation of the toy example and additional experimental results.
- More comprehensive evaluation and ablation study.
- User study.
- Further implementation details of the experiments.
- Detailed Prompts for the Experiments.

**Appendix C** The third part briefly discusses future work and potential applications.

## A    EXTENDED DISCUSSION AND ANALYSIS OF OUR METHODS

### A.1    A PRINCIPLED DERIVATION OF NAIVE $S^2$-GUIDANCE FROM A BAYESIAN PERSPECTIVE

In this subsection, we provide a principled theoretical foundation for our proposed Naive Stochastic Sub-network Guidance (Naive $S^2$-Guidance) method. We move beyond a heuristic interpretation and formally derive our approach by drawing a direct line to the principles of Bayesian inference, as established in the seminal work "Dropout as a Bayesian Approximation" by Gal & Ghahramani (2016). Our central argument is that Naive $S^2$-Guidance is not merely inspired by Bayesian ideas, but can be derived as a principled mechanism for correcting the predictions of a deterministic model by leveraging its own epistemic uncertainty.

#### A.1.1    FOUNDATIONAL BAYESIAN FORMULATION

Let $\mathcal{D} = \{x_i, c_i\}_{i=1}^{M}$ be our training dataset. A fully Bayesian approach to generative modeling would seek to compute the true posterior predictive distribution for a new sample $x_t$ given a condition $c$:

$$p(D|x_t, c, \mathcal{D}) = \int_{\boldsymbol{\Theta}} p(D|x_t, c, \boldsymbol{\theta}) p(\boldsymbol{\theta}|\mathcal{D}) d\boldsymbol{\theta}, \tag{5}$$

where $\boldsymbol{\theta} \in \boldsymbol{\Theta}$ are the model parameters, $p(\boldsymbol{\theta}|\mathcal{D})$ is the true posterior distribution over these parameters, and $p(D|x_t, c, \boldsymbol{\theta})$ is the likelihood of a specific prediction $D$ given parameters $\boldsymbol{\theta}$. The true posterior is given by Bayes' theorem:

$$p(\boldsymbol{\theta}|\mathcal{D}) = \frac{p(\mathcal{D}|\boldsymbol{\theta})p(\boldsymbol{\theta})}{p(\mathcal{D})} = \frac{p(\mathcal{D}|\boldsymbol{\theta})p(\boldsymbol{\theta})}{\int_{\boldsymbol{\Theta}} p(\mathcal{D}|\boldsymbol{\theta}')p(\boldsymbol{\theta}')d\boldsymbol{\theta}'}. \tag{6}$$

The integral in the denominator, known as the marginal likelihood or model evidence, is intractable for deep neural networks. To circumvent this, we employ Variational Inference (VI), introducing a tractable approximate posterior distribution $q_\phi(\boldsymbol{\theta})$ (parameterized by $\phi$) to approximate $p(\boldsymbol{\theta}|\mathcal{D})$. We minimize the Kullback-Leibler (KL) divergence between these two distributions:

$$\phi^* = \arg\min_{\phi} \text{KL}(q_\phi(\boldsymbol{\theta})||p(\boldsymbol{\theta}|\mathcal{D})) \tag{7}$$

$$= \arg\min_{\phi} \int q_\phi(\boldsymbol{\theta}) \log \frac{q_\phi(\boldsymbol{\theta})}{p(\boldsymbol{\theta}|\mathcal{D})} d\boldsymbol{\theta} \tag{8}$$

$$= \arg\min_{\phi} \int q_\phi(\boldsymbol{\theta}) \log \frac{q_\phi(\boldsymbol{\theta})p(\mathcal{D})}{p(\mathcal{D}|\boldsymbol{\theta})p(\boldsymbol{\theta})} d\boldsymbol{\theta} \tag{9}$$

$$= \arg\min_{\phi} \left( \text{KL}(q_\phi(\boldsymbol{\theta})||p(\boldsymbol{\theta})) - \mathbb{E}_{q_\phi(\boldsymbol{\theta})}[\log p(\mathcal{D}|\boldsymbol{\theta})] \right). \tag{10}$$

Minimizing this objective is equivalent to maximizing the Evidence Lower Bound (ELBO), $\mathcal{L}_{\text{ELBO}}$. The work of Gal & Ghahramani (2016) provides the theoretical grounding for interpreting stochastic network perturbations, such as dropout, as a form of this Bayesian optimization.

In our work, we generalize this concept from neuron-level dropout to block-level dropout. Each binary mask $\mathbf{m}_i \sim p(\mathbf{m})$ applied via stochastic block dropping effectively samples a specific set of weights $\boldsymbol{\theta}_i = \boldsymbol{\theta} \odot \mathbf{m}_i$ from this approximate posterior, which we denote simply as $q(\boldsymbol{\theta})$.

#### A.1.2    MONTE CARLO ESTIMATION OF THE APPROXIMATE POSTERIOR PREDICTIVE

The prediction of a single sub-network, $\hat{D}_\theta(x_t \mid c, \mathbf{m}_i)$, is a legitimate sample from the *approximate posterior predictive distribution*, $p_q(D|x_t, c)$:

$$\hat{D}_\theta(x_t \mid c, \mathbf{m}_i) \triangleq D(x_t \mid c; \boldsymbol{\theta}_i), \quad \text{where} \quad \boldsymbol{\theta}_i \sim q(\boldsymbol{\theta}). \tag{11}$$

The first moment of this distribution, the posterior mean $\mu_{\text{post}}$, is theoretically defined as the integral over the variational distribution:

$$\mu_{\text{post}}(x_t \mid c) \triangleq \mathbb{E}_{q(\boldsymbol{\theta})}[D(x_t \mid c; \boldsymbol{\theta})] = \int D(x_t \mid c; \boldsymbol{\theta}) q(\boldsymbol{\theta}) d\boldsymbol{\theta}. \tag{12}$$

Since this integral is analytically intractable for deep neural networks, we rely on **Monte Carlo integration** to estimate it. The empirical average computed by our algorithm serves as this estimator:

$$\hat{\mu}_{\text{post}}(x_t \mid c) \approx \frac{1}{N} \sum_{i=1}^{N} \hat{D}_\theta(x_t \mid c, \mathbf{m}_i). \tag{13}$$

Computing high-dimensional integrals via empirical averaging over diverse predictive hypotheses is a standard practice in deep learning. This paradigm is supported by extensive literature, including explicit methods like Lakshminarayanan et al. (2017); Huang et al. (2017); Gal & Ghahramani (2016); Lei et al. (2023); Wen et al. (2020). These works collectively establish that aggregating predictions from stochastic sub-states or ensemble members effectively approximates the predictive posterior. Our algorithm is a direct application of this principle to the sub-networks induced by block-dropping.

This posterior mean, $\mu_{\text{post}}$, represents the "center of mass" of the model's belief. The second central moment, the variance, quantifies the **epistemic uncertainty**:

$$\text{Var}_{q(\boldsymbol{\theta})}[D(x_t \mid c; \boldsymbol{\theta})] = \mathbb{E}_{q(\boldsymbol{\theta})}\big[(D(x_t; \boldsymbol{\theta}) - \mu_{\text{post}})^2\big]$$

$$\approx \frac{1}{N} \sum_{i=1}^{N} \big(\hat{D}_\theta(x_t; \mathbf{m}_i) - \hat{\mu}_{\text{post}}(x_t)\big)^2. \tag{14}$$

Our central hypothesis is that **low-quality generative outputs often arise in regions of high epistemic uncertainty**. In such regions, the posterior mean, $\mu_{\text{post}}$, often corresponds to a "safe," but ultimately low-quality output (e.g., a blurry artifact). The deterministic MAP estimate, $D_\theta(x_t \mid c)$, however, might be unjustifiably confident in these very regions.

### A.1.3 Deriving $S^2$-Guidance as an Uncertainty-Aware Correction

Based on this hypothesis, we formulate a principled correction to the Classifier-free Guidance (CFG) prediction, $\tilde{D}_{\text{CFG}}$. The standard guidance is:

$$\tilde{D}_{\text{CFG}}(x_t \mid c) = D_\theta(x_t \mid \phi) + \lambda\big(D_\theta(x_t \mid c) - D_\theta(x_t \mid \phi)\big). \tag{15}$$

We define our corrected prediction, $\tilde{D}_\theta^{\lambda,\omega}(x_t \mid c)$, as the solution to an optimization problem where we seek a prediction that remains faithful to the original guidance while being repelled from the center of uncertainty. Let us define a correction vector $\Delta D$. We propose that this correction should be in the direction opposite to the posterior mean, which acts as the locus of uncertainty-induced artifacts:

$$\Delta D \triangleq -\omega \cdot \mu_{\text{post}}(x_t \mid c), \tag{16}$$

where $\omega$ is a scalar controlling the magnitude of the repulsion. The corrected prediction is thus the linear superposition of the original guidance and this correction term:

$$\tilde{D}_\theta^{\lambda,\omega}(x_t \mid c) \triangleq \tilde{D}_{\text{CFG}}(x_t \mid c) + \Delta D, \tag{17}$$

$$= \tilde{D}_{\text{CFG}}(x_t \mid c) - \omega \cdot \mu_{\text{post}}(x_t \mid c), \tag{18}$$

$$= \underbrace{D_\theta(x_t \mid \phi) + \lambda\big(D_\theta(x_t \mid c) - D_\theta(x_t \mid \phi)\big)}_{\text{Standard CFG}}$$

$$- \underbrace{\omega \cdot \mathbb{E}_{q(\boldsymbol{\theta})}[D(x_t \mid c; \boldsymbol{\theta})]}_{\text{Uncertainty-Aware Repulsion Term}}. \tag{19}$$

Substituting the Monte Carlo approximation from Eq. 13 into Eq. 17, we recover our full Naive $S^2$-Guidance formulation:

$$\tilde{D}_\theta^{\lambda,\omega}(x_t \mid c) = D_\theta(x_t \mid \phi) + \lambda\big(D_\theta(x_t \mid c) - D_\theta(x_t \mid \phi)\big)$$

$$- \frac{\omega}{N} \sum_{i=1}^{N} \hat{D}_\theta(x_t \mid c, \mathbf{m}_i). \tag{20}$$

### A.1.4 THEORETICAL INTERPRETATION AND DECOMPOSITIONS

This derivation provides a much deeper understanding of Naive $S^2$-Guidance.

**Decomposition of Predictive Components.** We can rearrange Eq. 19 to analyze the contribution of each component to the final prediction:

$$\tilde{D}_\theta^{\lambda,\omega}(x_t \mid c) = (1 - \lambda)D_\theta(x_t \mid \phi) + \lambda D_\theta(x_t \mid c)$$
$$- \omega \cdot \mu_{\text{post}}(x_t \mid c) \tag{21}$$
$$= \underbrace{\lambda D_\theta(x_t \mid c)}_{\text{MAP Guidance}} + \underbrace{(1 - \lambda)D_\theta(x_t \mid \phi)}_{\text{Unconditional Prior}}$$
$$- \underbrace{\omega \cdot \mu_{\text{post}}(x_t \mid c)}_{\text{Bayesian Correction}}. \tag{22}$$

This shows a clear trade-off: we leverage the strong guidance from the conditional MAP estimate ($D_\theta(x_t \mid c)$) and the unconditional prior ($D_\theta(x_t \mid \phi)$), but temper both with a Bayesian correction term that represents the consensus of a diverse committee of model hypotheses. It acts to regularize the overconfidence of the single MAP estimate.

**A Gradient-Space Perspective.** In diffusion models, the guidance is applied in the noise prediction space. Let $\epsilon_\theta(x_t, c)$ be the model's noise prediction. The standard CFG-guided noise $\tilde{\epsilon}_{\text{CFG}}$ is:

$$\tilde{\epsilon}_{\text{CFG}}(x_t, c) = \epsilon_\theta(x_t, \phi) + \lambda(\epsilon_\theta(x_t, c) - \epsilon_\theta(x_t, \phi)). \tag{23}$$

Our method introduces a correction term directly in this space. Let $\bar{\epsilon}_{\text{post}}(x_t, c) = \mathbb{E}_{q(\theta)}[\epsilon_\theta(x_t, c; \theta)]$ be the posterior mean of the noise prediction. Our corrected noise prediction becomes:

$$\tilde{\epsilon}_{S^2G}(x_t, c) = \tilde{\epsilon}_{\text{CFG}}(x_t, c) - \omega \cdot \bar{\epsilon}_{\text{post}}(x_t, c) \tag{24}$$

$$= \tilde{\epsilon}_{\text{CFG}}(x_t, c) - \omega \cdot \left( \frac{1}{N} \sum_{i=1}^{N} \epsilon_\theta(x_t, c; \theta_i) \right). \tag{25}$$

This reveals that Naive $S^2$-Guidance is performing a direct modification of the guidance vector at each step of the denoising process. The repulsion from the "center of uncertainty" is not an abstract concept but a concrete vector subtraction in the high-dimensional noise space.

**Connection to Negative Ensemble Distillation.** Our method can be framed as a novel form of *negative distillation* applied at inference time. Standard ensemble distillation trains a single model to mimic the average output of an ensemble. In contrast, Naive $S^2$-Guidance uses the ensemble's average prediction ($\mu_{\text{post}}$) not as a target to be imitated, but as an anti-target to be actively repelled. This "distillation-rejection" mechanism is a new and principled way to harness the wisdom of an ensemble without collapsing to its mean.

In summary, Naive $S^2$-Guidance is a theoretically grounded method that leverages the principles of Bayesian model averaging and uncertainty quantification. It operationalizes the insight that high-quality generation requires not only strong conditional guidance but also a mechanism to actively avoid regions of high model uncertainty. Our derivation shows that subtracting the Monte Carlo average of stochastic sub-networks is a direct and principled way to implement this avoidance, thereby correcting for the inherent limitations of a single, deterministic generative model, as shown in Figure 9.

### A.2 COMPARATIVE ANALYSIS OF $S^2$-GUIDANCE AND NAIVE $S^2$-GUIDANCE

Our investigation into the behavior of sub-networks reveals a crucial property. We find that when the stochastic block-dropping ratio is constrained within a specific range, the guidance provided by different sub-networks appears remarkably consistent. As illustrated in Figure 10, even when different blocks are dropped to form distinct sub-network configurations, their individual guidance effects on the model's output distribution exhibit a strong similarity.

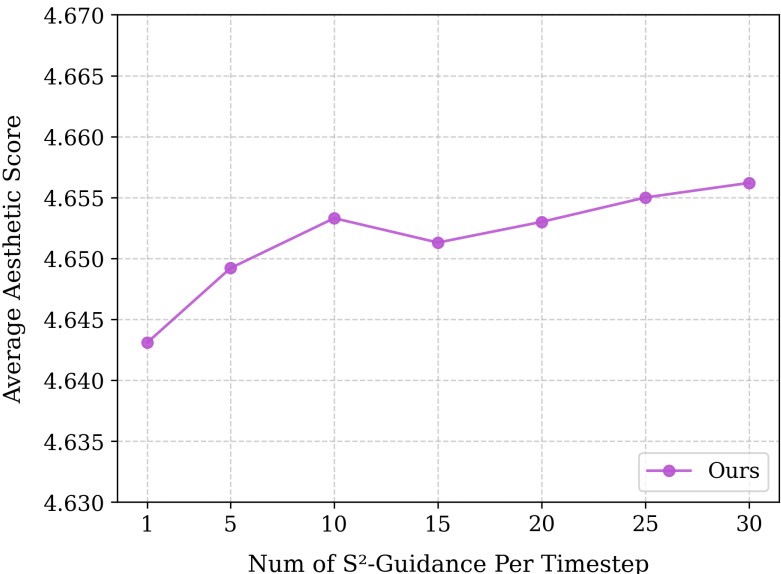

Figure 8: **Aesthetic score gains brought by increasing the number of forward passes with stochastic block dropping at each time step.**

This consistent behavior motivates us to formalize the relationship between the two methods using the principle of **unbiased estimation**.

Let $\theta$ be the model parameters and $p(\mathbf{m})$ be the distribution of binary masks. Following Gal & Ghahramani (2016), the stochastic block-dropping process induces a variational distribution $q(\tilde{\theta})$ over the parameter space. We define the **Theoretical Expected Guidance** (the population mean) as the exact predictive mean under this induced distribution:

$$\mathcal{G}^* \triangleq \omega \cdot \mathbb{E}_{q(\tilde{\theta})}[D(x_t \mid c; \tilde{\theta})] \equiv \omega \cdot \mathbb{E}_{\mathbf{m} \sim p(\mathbf{m})}[\hat{D}_\theta(x_t \mid c, \mathbf{m})]. \tag{26}$$

**Naive $S^2$-Guidance** approximates this target using a Monte Carlo average of $N$ i.i.d. samples:

$$G_{\text{Naive}} = \frac{\omega}{N} \sum_{i=1}^{N} \hat{D}_\theta(x_t \mid c, \mathbf{m}_i). \tag{27}$$

By the linearity of expectation, it holds that $\mathbb{E}[G_{\text{Naive}}] = \mathcal{G}^*$.

In contrast, our simplified $S^2$-**Guidance** employs a stochastic guidance term from a single sample ($N = 1$):

$$G_{S^2\text{-Guidance}} = \omega \cdot \hat{D}_\theta(x_t \mid c, \mathbf{m}_t), \quad \text{where} \quad \mathbf{m}_t \sim p(\mathbf{m}). \tag{28}$$

We formally derive that $G_{S^2\text{-Guidance}}$ is also an unbiased estimator of the same theoretical target $\mathcal{G}^*$:

$$\mathbb{E}_{p(\mathbf{m}_t)}[G_{S^2\text{-Guidance}}] = \mathbb{E}_{p(\mathbf{m}_t)}[\omega \cdot \hat{D}_\theta(x_t \mid c, \mathbf{m}_t)] = \mathcal{G}^*. \tag{29}$$

Since $\mathbb{E}[G_{S^2\text{-Guidance}}] = \mathbb{E}[G_{\text{Naive}}] = \mathcal{G}^*$, both methods are mathematically **unbiased Monte Carlo estimators** of the same target, differing only in variance. While $G_{S^2\text{-Guidance}}$ naturally exhibits higher variance per step compared to the ensemble average $G_{\text{Naive}}$, the iterative nature of diffusion sampling effectively performs temporal integration. This smooths out the stochastic noise over the trajectory (as confirmed by our variance analysis in Appendix B), confirming that a single stochastic sample is sufficient and theoretically justified.

Further experiments, such as repeating the process with a small number of samples (as shown in Figure 8), corroborate this perspective by demonstrating diminishing returns, validating the practical efficiency of our approach.

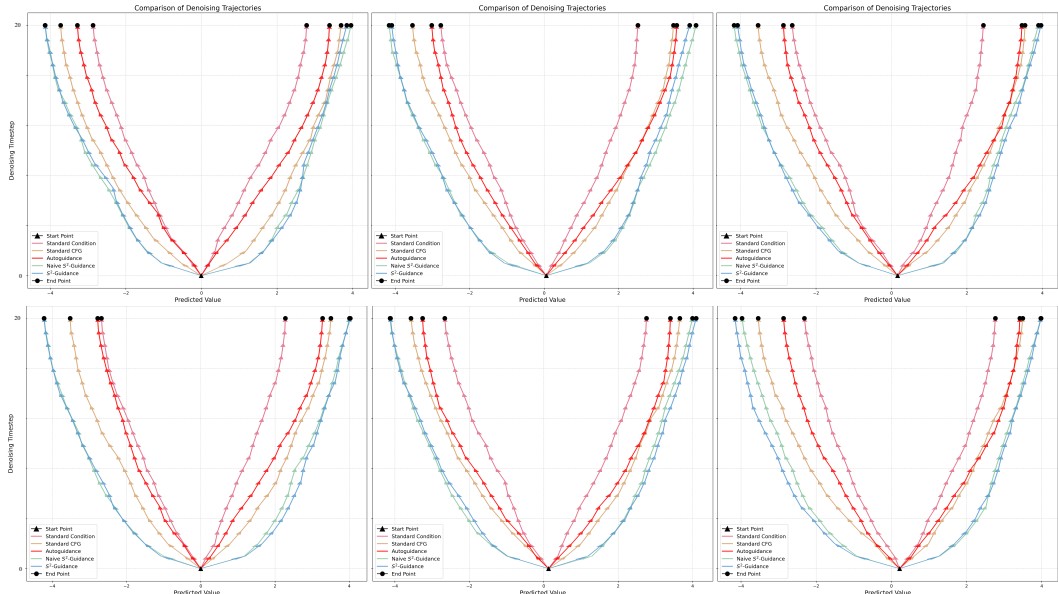

Figure 9: **Visualization of Denoising Trajectories on the 1D Bimodal Gaussian Data.** Each panel shows the paths taken by different guidance methods to generate samples targeting the ground truth modes at -4 and 4. The y-axis represents the denoising timestep (from start to end), and the x-axis shows the predicted sample value. While standard CFG and Autoguidance improve upon the unguided baseline, they consistently fail to reach the ground truth. In contrast, both **Naive S²-Guidance** and our final **S²-Guidance** method successfully steer the generation process to the correct endpoints. The more direct paths of our methods indicate a more accurate guidance signal throughout the entire denoising process.

## B  MORE DETAILS ABOUT OUR EXPERIMENTS

### B.1  TOY EXAMPLES

#### B.1.1  MORE RESULTS OF TOY EXAMPLES

To further analyze the guidance mechanisms, we visualize the full denoising trajectories for the 1D Bimodal Gaussian Distribution in Figure 9. The figure illustrates that while standard CFG and Autoguidance improve upon the unguided baseline, their final predictions consistently deviate from the distributions centered at -4 and 4. This visually demonstrates the mode-shifting problem discussed in the main paper.

In stark contrast, the paths for both Naive S²-Guidance and our final S²-Guidance method are more direct and successfully converge to the correct endpoints. This suggests that our self-guidance signal effectively corrects the generation path at each timestep, preventing the model from settling in the suboptimal regions favored by other methods.

#### B.1.2  NAIVE $S^2$-GUIDANCE VERSUS $S^2$-GUIDANCE IN TOY EXAMPLES

In our methodology, we first proposed Naive S²-Guidance, which averages the predictions from multiple stochastic sub-networks to create a robust negative guidance signal. However, this approach carries a significant computational cost. To address this, we introduced our final, more efficient S²-Guidance, which uses only a single stochastic sub-network per timestep.

To validate that this simplification does not cause a meaningful performance degradation, we conduct a direct comparison on the 2D Gaussian mixture. As illustrated in Figure 10, the sample distributions generated by both methods are qualitatively indistinguishable across multiple independent runs. Both approaches effectively guide the generation process to the correct modes and prevent the mode collapse issues seen in standard CFG (see Figure 3 in the main paper).

Given the negligible difference in performance, the substantial computational advantage of S²-Guidance makes it the far more practical and efficient choice than the naive variant. This finding strongly supports our adoption of the simplified approach as our final method.

### B.1.3 DETAILS OF TOY EXAMPLES

Below are the implementation details for the experiments on both synthetic and real-world data, as referenced in the main paper. All experiments were conducted using class-balanced datasets to assess the performance of our method.

- **1-D Bimodal Gaussian Distribution:** This experiment was designed to test the model's ability to stably and completely capture both modes of a bimodal distribution. The ground-truth data is an equally-weighted mixture of two Gaussians. The diffusion model, parameterized by a standard neural network, was trained for iterations to reconstruct the target distribution. Analysis involved visualizing the final sample distribution and denoising trajectories to show that $S^2$-Guidance consistently covers both modes, whereas the baseline may exhibit instability or mode preference (see Figure 3 in the main paper).

- **2-D Gaussian Mixture (4-Modes):** This experiment assessed the model's capacity to generate samples from a disconnected, multi-modal manifold. The data consisted of an equally-weighted mixture of 4 isotropic Gaussians, with means located at $(-4, -4)$, $(-4, 4)$, $(4, -4)$, and $(4, 4)$. The analysis focused on the final distribution and denoising paths to demonstrate that $S^2$-Guidance successfully captures all 4 distinct modes, improving upon the baseline's mode coverage.

- **Real-Image Data (CIFAR-10):** To validate $S^2$-Guidance on high-dimensional data, we used a class-balanced dataset from CIFAR-10, consisting of 5,000 **'horse'** images and 5,000 **'car'** images. The diffusion model employed a neural network parameterization common for image tasks. The primary goal of the analysis was to assess the quality and class-separability of the generations. To this end, we generated 3,000 images and visualized their **CLIP (ViT-B/32) features** in 2-D using **t-SNE**. The resulting plot demonstrates that $S^2$-Guidance produces more distinct and well-separated class clusters compared to the baseline, indicating higher-quality and less ambiguous generations (see Figure 4 in the main paper).

### B.2 EXTENDED EVALUATIONS

**Experiments using Flux.** In addition to the main experiments conducted with SD3 and SD3.5, we further evaluate our method using Flux (Labs, 2024), a state-of-the-art (SOTA) model for text-to-image generation. Note that Flux is a CFG-distilled model, meaning that directly applying classifier-free guidance (CFG) may lead to different results. We use a De-distilled version of Flux (Labs, 2024) in our experiments. Additionally, we follow the same benchmark setting as HPSv2.1 to ensure consistency and comparability.

| Method | HPSv2.1(%) ↑ | | | | | Qalign↑ |
|---|---|---|---|---|---|---|
| | Anime | Concept Art | Paintings | Photo | Avg. | |
| CFG | 31.29 | 29.85 | 30.03 | 28.16 | 29.84 | 4.65 |
| CFG (1.4× NFE) | **31.59** | 30.10 | 30.35 | 28.47 | 30.13 | 4.68 |
| **Ours** | 31.48 | **30.21** | **30.48** | **28.88** | **30.26** | **4.70** |

Table 6: **Quantitative evaluation of CFG and our approach using Flux under the HPSv2.1 benchmark.** The HPSv2.1 grouping evaluates different styles, while Qalign measures aesthetic quality. Higher scores (↑) are better. Best results are in bold.

The results in Table 6 show that our method consistently outperforms the baseline CFG across different categories, including Anime, Concept Art, Paintings, and Photo. Specifically, we observe an average improvement of **0.42**, highlighting the robustness and effectiveness of our approach.

For more qualitative results, please refer to Figure 12 and Figure 13. These comprehensive results demonstrate the effectiveness of our proposed approach across various scenarios.

**Analysis of Variance from Stochastic Dropping**    To assess the stability of our method, we quantify the output variance introduced by the stochastic dropping of network blocks. In our experiment, we generate multiple images for the same prompt while keeping the initial noise seed fixed, thereby isolating the variance attributable solely to the stochastic dropping process. The quantitative results, presented in Table 7, demonstrate that the run-to-run variance is negligible. As shown, $S^2$-Guidance exhibits a variance on the order of $10^{-6}$ and a coefficient of variation of less than 1%. This indicates an extremely high degree of stability and output consistency, confirming that the stochastic element does not compromise the reliability of the generation process.

| Method | Mean (%) | Var. | Std. Dev. | Coeff. of Var. |
|---|---|---|---|---|
| CFG | 30.48 | – | – | – |
| $S^2$-Guidance | 30.86 | $7 \times 10^{-6}$ | 0.0026 | 0.84% |

Table 7: Analysis of variance from stochastic dropping with a fixed initial seed.

## B.3    USER STUDY

To quantitatively evaluate the perceptual quality and prompt fidelity of our method, we conducted a comprehensive user study comparing $S^2$-Guidance against four strong baselines: CFG (Ho & Salimans, 2022), APG (Sadat et al., 2024), CFG++ (Chung et al., 2024), and CFG-Zero (Fan et al., 2025). The evaluation was performed on images generated from a diverse set of diffusion models to assess the generalizability of our approach.

We recruited 14 participants with expertise in computer vision and generative AI. For each evaluation instance, participants were presented with a text prompt and the corresponding images generated by all five methods, displayed in a randomized order to prevent bias. Participants were instructed to evaluate the results based on three key criteria:

- **Detail Preservation:** The clarity, sharpness, and richness of details in the generated image.
- **Color Consistency:** The naturalness, harmony, and realism of the colors.
- **Image-Text Alignment:** How well the generated image accurately reflects the content and intent of the text prompt.

For each criterion, participants were asked to select the image (or images) they found to be the most successful. This design allows for multiple selections if a participant deems more than one result to be of high quality for a given aspect, thereby capturing a more nuanced assessment of performance.

The results of the user study are presented in Figure 11. The findings demonstrate a clear and consistent preference for our proposed method, $S^2$-Guidance, across all evaluated metrics. Specifically, in the *Detail Preservation* category, $S^2$-Guidance was preferred in 32.5% of cases, significantly outperforming the runner-up, CFG (18.3%). A similar dominant trend is observed for *Color Consistency*, where $S^2$-Guidance achieved a 29.6% preference rate. Furthermore, for *Image-Text Alignment*, our method was chosen 31.1% of the time, again marking a substantial lead over all baselines.

Aggregating the votes, the *Overall* preference for $S^2$-Guidance stands at 31.0%, confirming its comprehensive superiority. This strong performance in human evaluations validates that $S^2$-Guidance not only improves guidance from a theoretical standpoint but also translates to tangible and perceptually superior generation quality that is easily recognized by human users.

## B.4    IMPLEMENTATION DETAILS

To ensure fair comparisons, the implementation details of our experiments are as follows: For the text-to-image comparisons, we used SD3 and SD3.5 (Esser et al., 2024; AI, 2024) with the guidance scale set to 7.5. For our scale parameter $\omega$, we set it to 0.25. For the text-to-video comparisons, we use a guidance scale of 5.0. Similarly, our scale parameter $\omega$ is set to 0.25. All other hyperparameters are set to the default configurations of the respective models. For the baseline comparisons, we follow the original implementations provided in their official repositories. Specifically, APG (Sadat et al., 2024) and CFG++ (Chung et al., 2024) are implemented using the community-contributed

versions that are integrated into the Diffusers framework. All experiments are conducted on NVIDIA H20 GPUs with 96GB memory.

### B.5 DETAILED PROMPTS FOR FIGURE 1

This section provides the prompts used to generate the visual results presented in Figure 1. The examples are referenced by their grid position in the figure (row, column).

- **(Top, 1) Astronaut in space (Video):** *"An astronaut flying in space."*
- **(Top, 2) Floating Castle (Image):** *"A magnificent castle sitting high on a floating island above the clouds. Fluffy clouds surround the base of the island and form the text 'S2 Guidance Is All You Need' in a romantic, swirling style. The castle is adorned with towers, golden lights twinkling in the windows, and vines of blooming flowers climbing its walls. The scene is lit by a warm, golden light glowing from the sun, with a starry heaven faintly visible on the horizon."*
- **(Top, 3) Abstract Portrait (Image):** *"The bold dramatic strokes of the painter's brush created a stunning abstract masterpiece a work of emotional depth and intensity."*
- **(Top, 4) Cat with Rocket (Image):** *"A cat sitting besides a rocket on a planet with a lot of cactuses."*
- **(Top, 5) Sports Car Driving (Video):** *"a car accelerating to gain speed."*
- **(Bottom, 1) Woman with Colored Powder (Video):** *"A close-up of a beautiful woman's face with colored powder exploding around her, creating an abstract splash of vibrant hues."*
- **(Bottom, 2) Woman with Umbrella (Image):** *"A woman sitting under an umbrella in the middle of a restaurant."*
- **(Bottom, 3) Man Running on Beach (Image):** *"A man is running his hand over a smooth rock at the beach."*
- **(Bottom, 4) Clay Sheep (Image):** *"a red book and an ivory sheep."*
- **(Bottom, 5) Bear Climbing Tree (Video):** *"a bear climbing a tree."*

## C FUTURE WORK

Our method is training-free and plug-and-play, requiring no additional fine-tuning or retraining of the underlying model. This enables seamless integration into existing pipelines and allows practitioners to improve generation quality immediately at inference time. A promising direction is to further explore and generalize the proposed self-guidance mechanism, which is designed to mitigate uncertainty regions within the model and thereby correct suboptimal trajectories. This capability suggests broad applicability to tasks where small deviations can lead to noticeable quality degradation. In the image domain, more reliable internal guidance may improve (i) image or video editing (Zhu et al., 2024; Wang et al., 2024a;b) accuracy, where faithful execution of edit instructions is critical; (ii) human-preference alignment objectives (Chen et al., 2025a; Ma et al., 2024c; Huang et al., 2025a); and (iii) high-fidelity image synthesis (Lei et al., 2025; Chen et al., 2025b) under challenging prompts. In the video domain, the same principle may translate to improved motion coherence and temporal smoothness, potentially reducing temporal artifacts (Wang et al.; Mao et al., 2025; Su et al., 2025; Wu et al., 2025b; Ling et al., 2025; Feng et al., 2025a; Zhu et al., 2026; Ma et al., 2025b). Finally, our observations are grounded in the redundancy of transformer architectures, suggesting potential applicability beyond DiT architectures. An important future direction is to investigate whether analogous self-guidance mechanisms can improve faithfulness, robustness in LLMs and MLLMs, as well as enable broader potential applications (Yao et al., 2025; Yu et al., 2025; Xie et al., 2024; Wang et al., 2025; 2026; Wei et al., 2023; Feng et al., 2025b; Fang et al., 2024; 2026; Jiang et al., 2025; Xie et al., 2026; Huang et al., 2025b; Chu et al., 2025a; 2021a;b; Huang et al., 2025c).

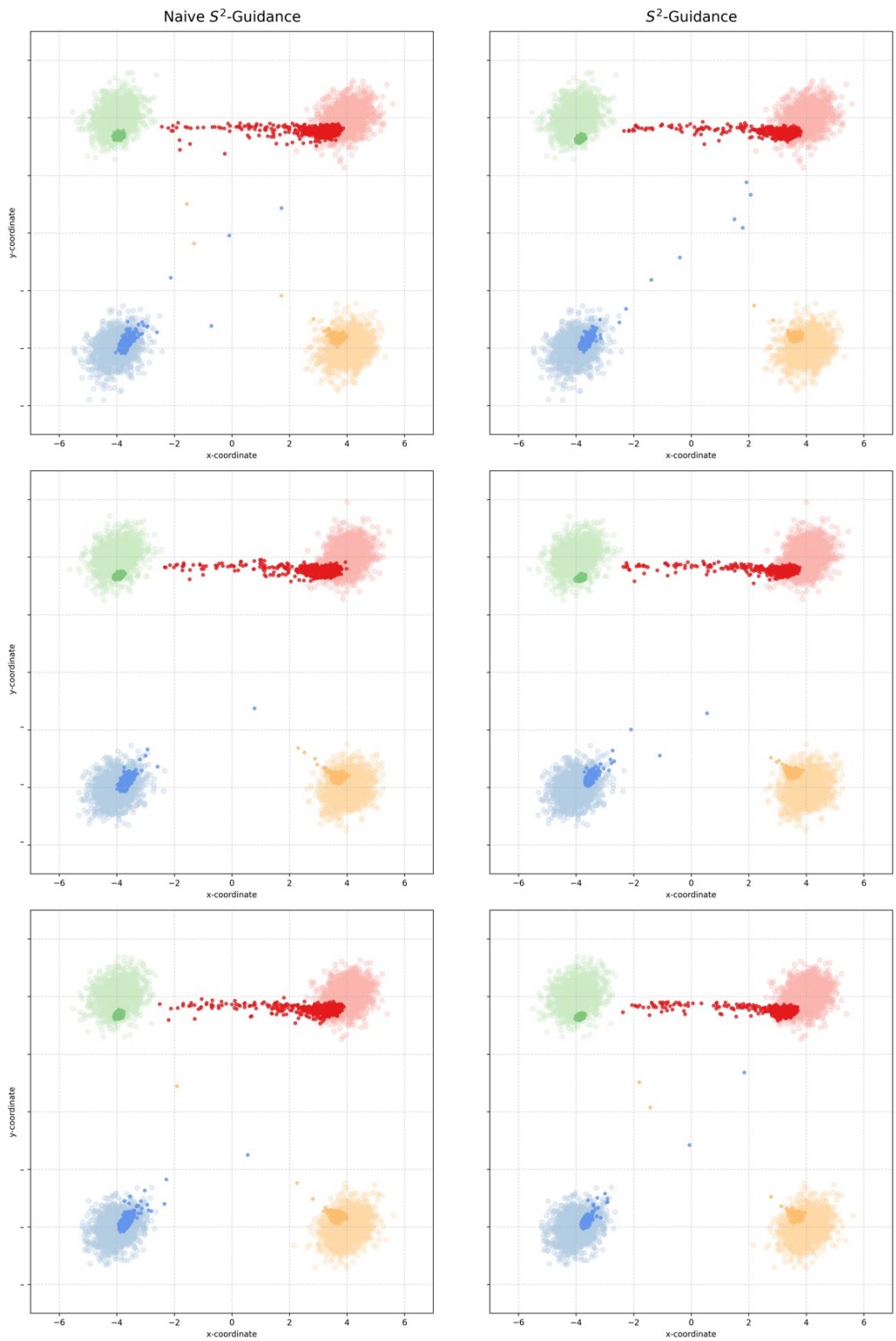

Figure 10: **More Visual Comparisons of Naive S²-Guidance and S²-Guidance on the 2D Gaussian Mixture.** Left: Naive S²-Guidance. Right: S²-Guidance. Each row corresponds to a different random seed. The generated sample distributions are virtually identical, demonstrating that the performance gain from the computationally intensive naive approach is minimal. This justifies our adoption of the more efficient S²-Guidance method.

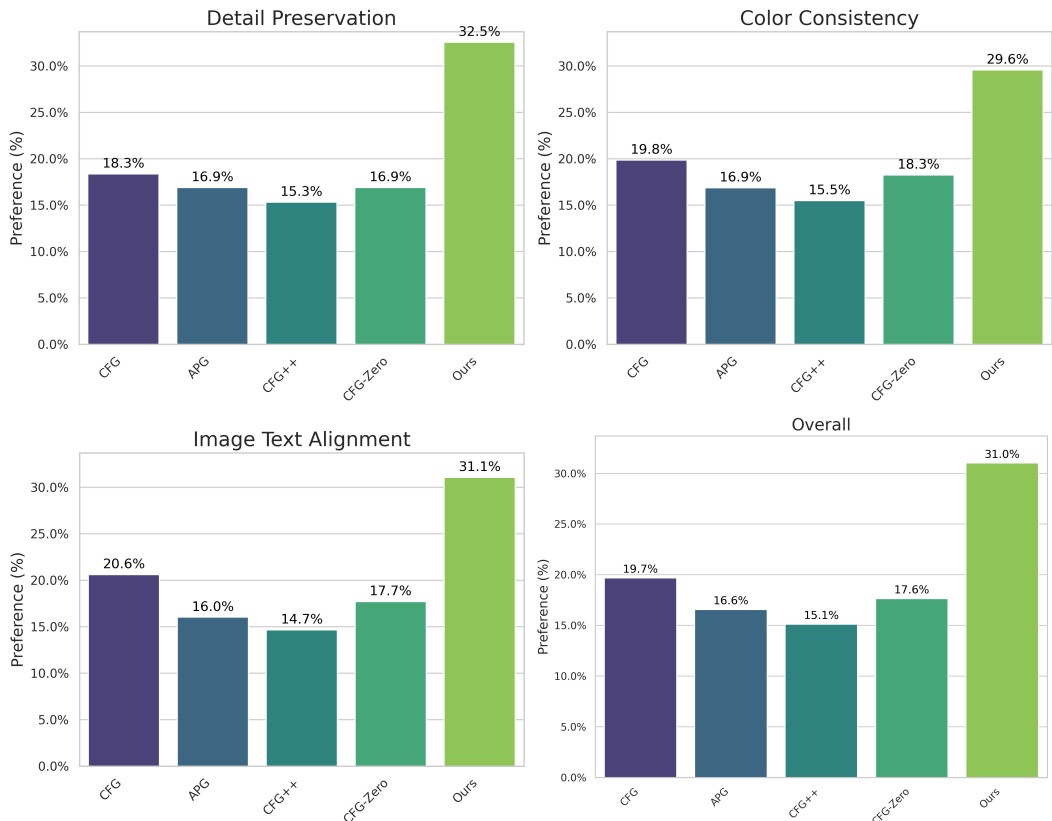

Figure 11: **Human preference evaluation results for $S^2$-Guidance against baseline methods.** The bar charts show the percentage of times each method was selected as the best for three criteria: Detail Preservation, Color Consistency, and Image-Text Alignment, along with an Overall aggregated score. Our method, $S^2$-Guidance, is significantly preferred by human evaluators across all categories, achieving a preference rate of over 29% in every dimension and surpassing 30% overall. This demonstrates its robust ability to generate perceptually superior images.

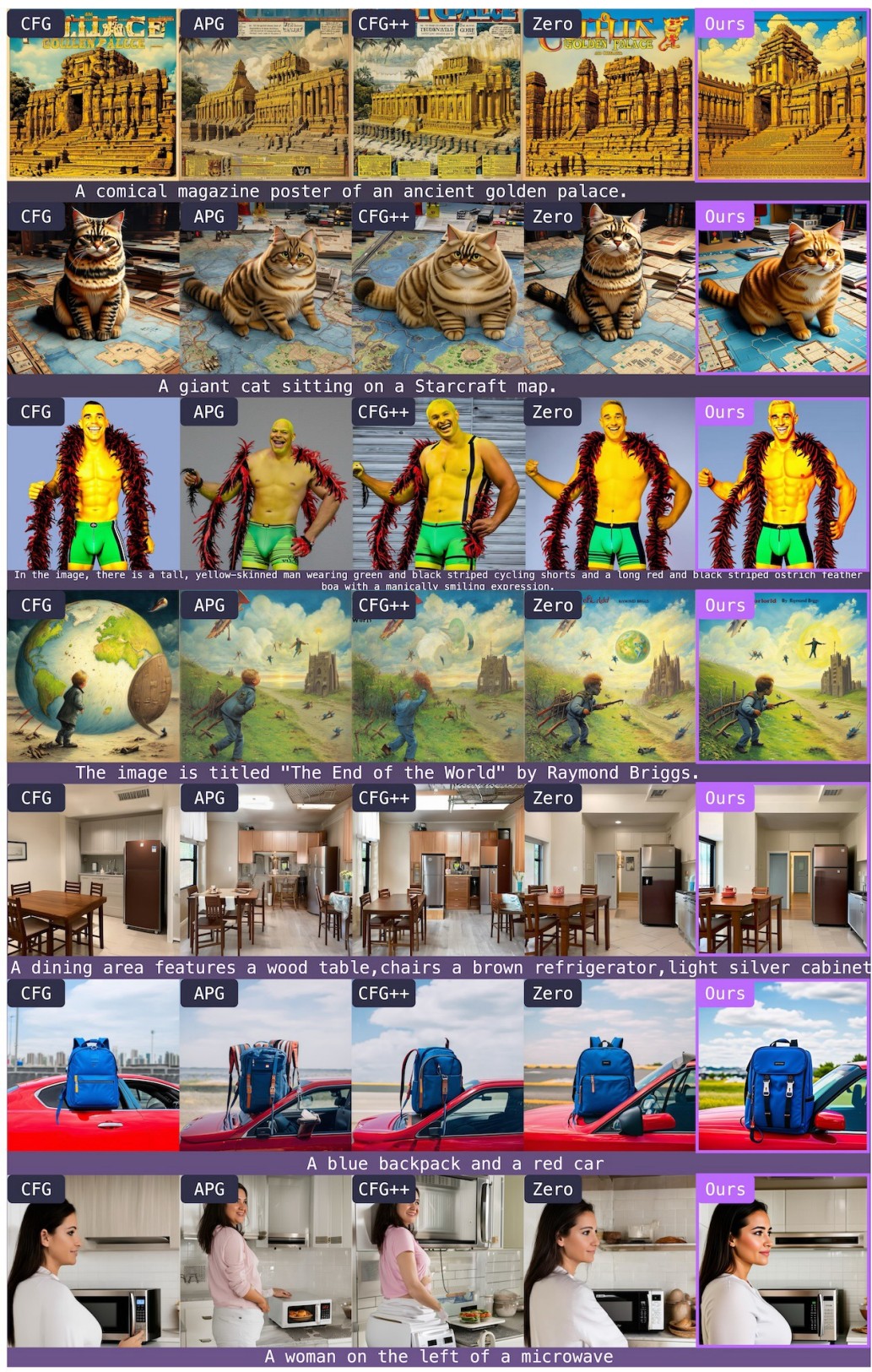

Figure 12: **Qualitative comparison of $S^2$-Guidance with baseline methods.** Our method consistently generates images with superior visual quality, better prompt alignment, and fewer artifacts across a variety of prompts. For instance, $S^2$-Guidance excels at stylistic replication (row 4), complex concept combinations (row 5).

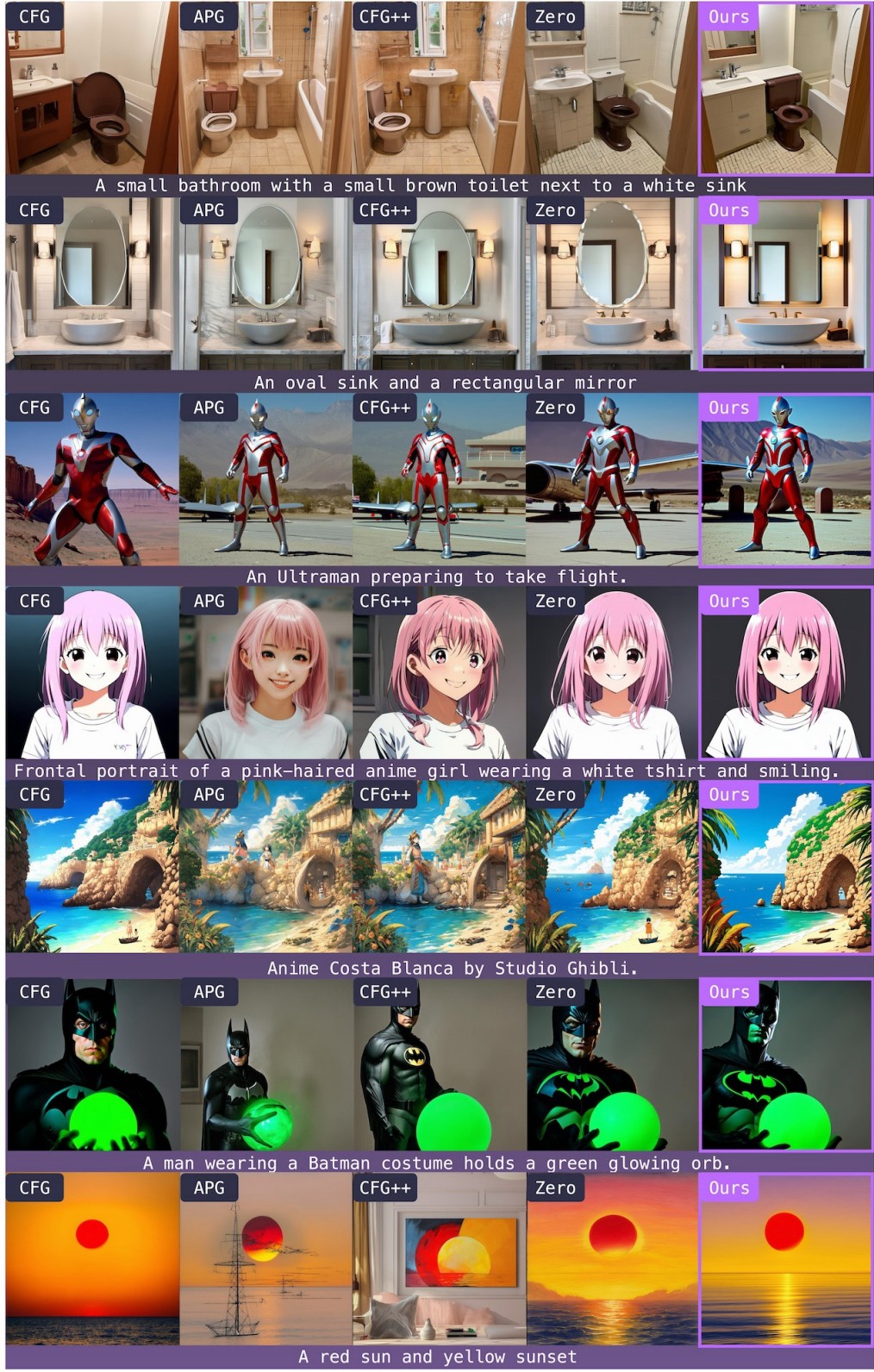

Figure 13: **Further qualitative comparisons.** Our approach demonstrates robust improvements in both prompt fidelity and aesthetic quality. Key advantages include accurate attribute binding (e.g., "oval sink and rectangular mirror" in row 2), faithful character and style generation (rows 3-5), and superior handling of lighting and composition (rows 6, 7). $S^2$-Guidance consistently avoids the conceptual blending and visual artifacts that affect other methods.

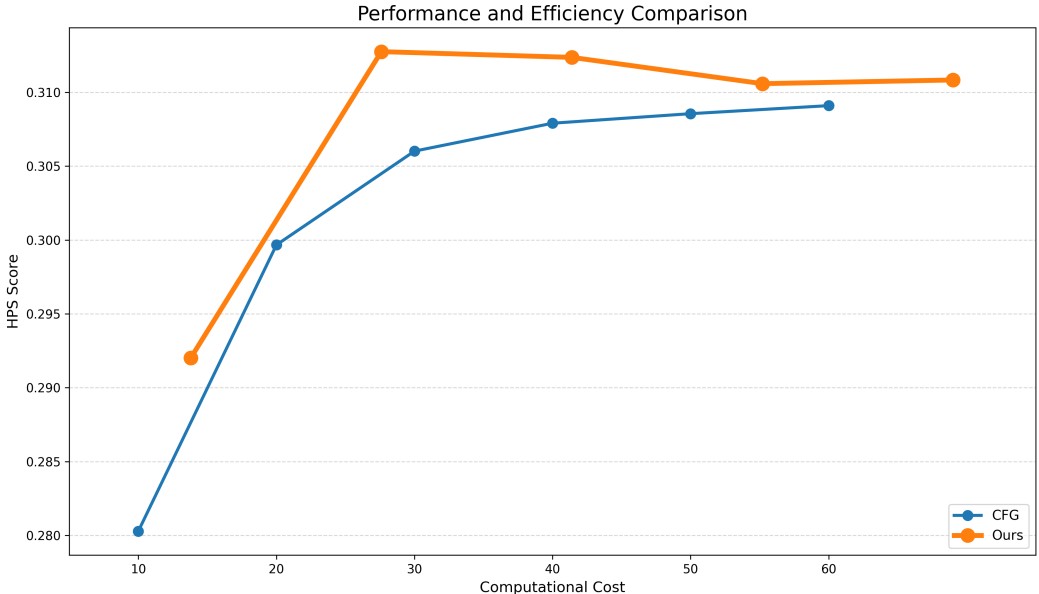

Figure 14: **Performance-Efficiency Trade-off Analysis.** This figure compares our method against CFG by plotting HPS Score as a function of computational cost. (Curves positioned further toward the **top-left** indicate superior methods.) The x-axis represents a normalized computational cost, where the cost for CFG equals its inference steps, while the cost for our method is scaled by a factor of 1.4 to reflect its ∼40% computational overhead. The plot illustrates that our method offers a significantly better trade-off. For instance, our method with just 20 inference steps (equivalent cost ≈ 28) already achieves a higher HPS score than CFG at 60 steps. This demonstrates that our method yields substantial quality improvements for a comparable or even lower computational budget.

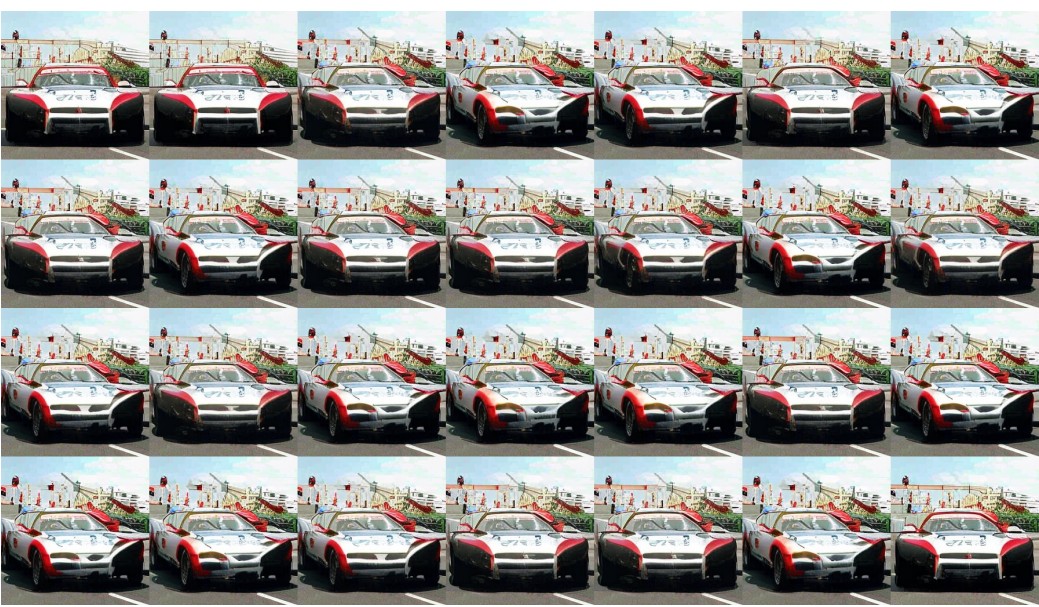

Figure 15: **Impact of dropping a single, fixed transformer block in SiT-XL.** Each of the 28 images corresponds to dropping one specific block for all timesteps on the ImageNet 256×256 task. The visual consistency across the grid demonstrates the model's robustness against block-level perturbations.

