# OpenReview forum: "Stochastic Self-Guidance for Training-Free Enhancement of Diffusion Models"
_ICLR.cc/2026/Conference — ICLR 2026 Poster_

### Official Review · Reviewer_PEh4 · 2025-10-26

**Soundness:** 3
**Presentation:** 3
**Contribution:** 3
**Rating:** 6
**Confidence:** 4

**Summary:**

This paper proposes a general and elegant strategy to enhance generative performance by mitigating issues of low fidelity and semantic incoherence. Specifically, the authors empirically demonstrate that suboptimal predictions of diffusion models can be effectively rectified using the model’s own sub-networks, without requiring additional training or the integration of external modules. Building upon this observation, the paper introduces S²-Guidance, a novel method that leverages stochastic block-dropping during the denoising process to activate sub-networks for self-guidance. The experiments conducted on three tasks show the great performance of the proposed method.

**Strengths:**

1. Clear and Insightful Motivation. The paper identifies a fundamental limitation in classifier-free guidance (CFG)—its tendency to over-amplify conditional signals and distort the underlying data distribution. The authors’ analysis of this phenomenon using Gaussian mixture toy examples is both illustrative and rigorous. This strong diagnostic foundation naturally motivates the development of S²-Guidance as a self-correcting mechanism.

2. Elegant and Minimalist Design. The proposed method achieves a notable balance between simplicity and effectiveness. By leveraging stochastic block dropping within the model itself, S²-Guidance creates intrinsic “weak sub-networks” that serve as self-guiding components. This design eliminates the need for externally trained weak models or additional data, while maintaining competitive performance and extremely low computational overhead. The minimal intervention required for implementation highlights the method’s engineering practicality and reproducibility.

3. Strong Empirical Validation and Robust Generalization. The experiments convincingly demonstrate that S²-Guidance enhances both sample fidelity and distributional consistency across diverse models and tasks. The method consistently improves over CFG and performs on par with, or better than, external weak-model guidance approaches—all without additional training or tuning. The results underscore its robustness and broad applicability to various diffusion architectures.

4. Well-Written and Structured Presentation.

**Weaknesses:**

1. Although the proposed method is simple yet efficient, its novelty remains limited. The authors introduce a more general form of the “weak” model; however, it still primarily targets essential problem-solving within existing frameworks rather than presenting a fundamentally new paradigm.

2. While the inclusion of the additional sub-optional model contributes to performance improvement, it also introduces extra computational cost and resource consumption.

3. The improvements in the text-to-video experiments are primarily reflected in enhanced visual quality. However, as shown in Table 4, the temporal consistency of the generated videos slightly degrades—particularly in terms of subject and background consistency. This suggests that the proposed method may offer limited benefits for video generation while demonstrating stronger advantages in image-level synthesis.

**Questions:**

None

---

> ### Author Response · Authors · 2025-11-24
> **Response to Reviewer PEh4 (1/2)**
>
> We sincerely thank you for your constructive comments and for your recognition of our work's strengths, including its *Clear and Insightful Motivation*, *Elegant and Minimalist Design*, *Strong Empirical Validation*, and *Well-Written and Structured Presentation*. Below, we provide a point-by-point response to the weaknesses you raised.
>
> ---
>
> ## W1: Method Novelty
>
> Thank you for recognizing our method as *simple yet efficient*. We believe that achieving substantial, consistent gains through simplicity and elegance is, in itself, a powerful form of innovation. We highlight our novelty from the following perspectives:
>
> *   **A Novel Solution to the "Weak Model" Bottleneck**
>
>     Our work represents **a pioneering effort** by introducing a novel method to source an effective "weak model" directly from the main model's intrinsic structure at inference time. This **training-free**, **plug-and-play** approach resolves the central bottleneck of prior guidance methods, which have been limited by *the practical infeasibility of training weak models for large-scale systems* and *the fragility of hand-crafted heuristics*. **By solving this bottleneck, S$^2$-Guidance transforms a promising guidance concept into a simple yet powerful mechanism**.
>
> *   **A Significant and Non-Trivial Empirical Innovation**
>
>     The power of this new mechanism is validated by its broad and consistent empirical impact. Our method delivers significant improvements across **6 major models**, **3 distinct tasks** (class-conditional, T2I, T2V), and **4 standard benchmarks** (HPSv2.1, T2I-CompBench, VBench, ImageNet). **Achieving such universally positive results with a single, elegant mechanism constitutes a significant empirical innovation**.
>
> *   **A Practical Contribution to the Community**
>
>     S²-Guidance offers immediate practical value as a **training-free** module. Its **seamless integration** allows easy enhancement of existing models without costly retraining, making advanced generative capabilities more accessible to the entire community.
>
> ---
>
> ## W2: Computational Cost and Efficiency
>
> Our design ensures this additional step is justified by **a more effective use of computational resources**, a claim supported by the following analyses:
>
> *   **Superior Performance at an Equivalent Computational Budget**
>
>     As shown in our **Table 4 (Appendix)**, when the total number of forward passes is kept constant, our method **consistently achieves higher scores** on both human preference (HPSv2.1) and aesthetic quality (QAlign) benchmarks. This demonstrates that for a **fixed computational budget**, our method yields a **superior result**.
>
> *   **New Systematic Trade-off Analysis**
>
>     To further address this concern, we have added a new performance-vs-efficiency analysis in **Figure 14 (Appendix)**. This plot illustrates that our method establishes a **more favorable performance-efficiency frontier**. S²-Guidance consistently occupies the **top-left region**, indicating it achieves a **higher level of performance for any given level of computational expenditure** compared to the alternatives.
>
> In conclusion, these analyses compellingly demonstrate our method's superiority by delivering higher quality for a given computational budget, establishing it as a more advanced and practical choice for maximizing generation quality.

---

> ### Author Response · Authors · 2025-11-24
> **Response to Reviewer PEh4 (2/2)**
>
> ## W3: T2V Performance and Temporal Consistency
>
> We would like to clarify that the slight decrease in specific consistency metrics is best understood as a direct consequence of a well-documented, inherent **trade-off** in video generation, **as noted by VBench itself**. Our results demonstrate that S$^2$-Guidance excels at navigating this trade-off to achieve a superior overall performance.
>
> *   **A Highly Favorable Trade-off in Our Results**
>
>     The VBench paper itself provides the direct rationale for this behavior. It highlights a fundamental trade-off between **Dynamic Degree** and **Temporal Consistency**, noting that models often achieve high consistency scores by "cheating" with "relatively static scenes." Conversely, creating videos with "large motions" is identified as the "current challenge" for maintaining consistency.
>
>     Our method demonstrates a highly advantageous position on this trade-off spectrum, achieving superior overall scores while significantly boosting dynamism, as summarized below:
>
>     | Model     | Method | Total($\uparrow$) | Quality ($\uparrow$) | Semantic ($\uparrow$) | Sub. Consis. ($\uparrow$) | Bg. Consis. ($\uparrow$) | Temp. Flick. ($\uparrow$) | Dyn. Degree ($\uparrow$)   |
>     | :-------- | :----- | :-------- | :---------- | :----------- | :--------------- | :--------------- | :--------------- | :---------------- |
>     | **Wan-1.3B**| CFG    | 80.29     | 84.32       | 64.16        | 96.53            | 95.46            | 99.49            | 73.61            |
>     |           | **Ours** | **80.93** | **84.74**   | **65.70**    | **96.57**        | **95.80**        | **99.52**        | **79.17 (+7.6%)**|
>     | **Wan-14B**| CFG    | 82.65     | 84.88       | 73.76        | 94.45            | 97.66            | 98.87            | 63.89             |
>     |           | **Ours** | **82.84** | **84.89**   | **74.65**    | 94.21            | 97.56            | **98.93**        | **65.28 (+2.2%)** |
>
>     These results clearly show:
>     *   On **Wan-1.3B**, our method achieves a massive **+7.6% gain in Dynamic Degree** while simultaneously improving across all other metrics, including a notable increase in the **Total Score** from 80.29 to **80.93**.
>     *   On **Wan-14B**, a substantial **+2.2% boost in Dynamic Degree** is achieved alongside **higher Total, Quality, and Semantic scores**, with only a negligible decrease (<0.3%) in two consistency sub-metrics.
>
>     This demonstrates that our method successfully tackles the core challenge of creating dynamic content rather than taking the easier path of generating static videos, while maintaining excellent overall performance.
>
> *   **Superior Overall Performance**
>
>     Crucially, by mastering this trade-off so effectively, our method achieves a **significantly higher Total Score** on VBench. Notably, the Quality and Semantic scores are also consistently higher. Specifically, our method scores **80.93 on Wan-1.3B** (vs. 80.29 for CFG) and **82.84 on Wan-14B** (vs. 82.65 for CFG). This strong overall performance helps to address the concern that our method's benefits might be "limited" for video generation. On the contrary, by intelligently navigating the task's inherent challenges, it delivers a stronger and more balanced overall performance, validating its effectiveness for text-to-video synthesis.
>
> ---
>
> Please let us know if any concerns remain unaddressed; we are happy to discuss them.

---

### Official Review · Reviewer_2Qir · 2025-10-31

**Soundness:** 3
**Presentation:** 3
**Contribution:** 2
**Rating:** 6
**Confidence:** 3

**Summary:**

This paper extends the idea of autoguidance, which generates negative predictions via stochastic dropout and improves sampling through extrapolation, by introducing block dropout to form negative predictions and formulate guidance. While selectively dropping certain blocks improved performance, it introduced a block selection problem, determining which blocks to drop. The authors address this issue by applying stochastic dropout during the sampling process, proposing S2-Guidance, and demonstrate improved performance on both GMM toy examples and real datasets.

**Strengths:**

- The comparison and visualization of different guidance methods on GMM data are clear and well-presented.
- Using block dropout instead of parameter dropout contributes to inference efficiency, akin to structured pruning, compared to conventional dropout.
- The observation that the specific choice of dropout block has limited impact on performance is important.
- Replacing manual block selection with stochastic dropout simplifies usage while maintaining or even improving general performance, making the approach more practical.

**Weaknesses:**

There are several concerns regarding the use of block dropout. Dropping entire model blocks could severely degrade prediction quality if applied to critical blocks, potentially corrupting the model’s behavior. While AutoGuidance adopts parameter-level dropout, this paper employs the stronger perturbation of block-level dropout without providing sufficient motivation or references for this design choice.

Indeed, as mentioned around L250-L254, applying naive S2-Guidance to key blocks drastically reduces model performance, suggesting that the block dropout mechanism’s impact is under-analyzed. Providing a visualization of how predictions change when each block is dropped would substantially strengthen the motivation and understanding of this method.

Furthermore, in L161-, the paper claims to offer a more general solution compared to previous weak model guidance methods, which relied on hand-crafted designs. However, since the proposed method still requires optimizing the dropout probability and potentially identifying key blocks to avoid, it may also involve a certain level of manual tuning, making the claim of being fully general somewhat overstated.

Although S2-Guidance removes explicit block selection by introducing stochastic dropout per timestep, the paper does not discuss whether dropping key blocks during random selection leads to performance degradation or whether the method produces consistent performance across different runs. A discussion or variance analysis on this point would be valuable. In addition, since this method perturbs model weights, the paper should further clarify the advantages and motivation of block-level dropout over the stochastic parameter dropout used in AutoGuidance.

Finally, the approach resembles PAG (Ahn et al., 2024), which applies attention masking to generate degraded samples and extrapolate them for guidance. It would be beneficial to include a comparison with PAG, as such a baseline could reinforce the paper’s empirical claims.

Minor

- In L158, the description of Hong et al. (2023) is inaccurate: the method performs attention-guided blurring of predicted samples, not blurring of attention maps themselves.

**Questions:**

- In Eq. (3), the guidance is formulated by subtracting the ω-scaled negative prediction, differing from the standard extrapolation-based CFG formulation.
- What is the rationale behind this design? Would applying similar ω-scaled subtraction to other bad-version guidance methods yield comparable effects?

---

> ### Author Response · Authors · 2025-11-23
> **Response to Reviewer 2Qir (1/4)**
>
> We sincerely thank you for your constructive comments and for your recognition of our work's strengths, including the *clear visualization* , *inference efficiency*, and the *practicality* of our approach. Below, we provide a point-by-point response to the weaknesses and questions you raised.
>
> ---
>
> ## W1 (the first paragraph): Motivation and Robustness of Block-Level Dropout
>
> The use of block-level dropout is a deliberate design choice. We clarify its *rationale* and address the concern of it being a *"strong perturbation"* by detailing: (1) the **motivation** from the limitations of prior work, (2) the **theoretical basis (references)** from architectural redundancy and established dropout mechanisms, and (3) the **empirical robustness** of block-level dropout.
>
> ### (1) Motivation: Overcoming the Limitations of Prior Work
> Our approach is directly **motivated by the practical scalability issues** of prior methods. AutoGuidance (*Karras et al., 2024*) relies on a separately trained "weak" model, but for modern large-scale pretrained models, acquiring such a specific degraded version (e.g., a version trained for fewer epochs) is often **infeasible**. Furthermore, as noted in AutoGuidance itself, an improperly chosen weak model can **fail to provide effective guidance**. This **necessitated a training-free, more practical approach** that does not rely on external auxiliary models.
>
> ### (2) Theoretical Basis (references): Drawing from Architectural Redundancy and Established Dropping Mechanisms
> Our theoretical basis is twofold. On the one hand, prior works (*Avrahami et al., 2025; Lou et al., 2024*) demonstrate that architectures like DiT exhibit significant redundancy, with high similarity across different transformer block outputs. **This inherent redundancy makes the architecture robust to the removal of some blocks.** On the other hand, **the mechanism of stochastically dropping is a well-established concept** in deep learning, validated by seminal works like *Stochastic Depth (Huang et al., 2016)* and given theoretical interpretation by methods like *Bayesian Dropout (Gal & Ghahramani, 2016)*. **The combination of an architecture known to be redundant with a well-established mechanism for stochastic dropping provides a strong rationale for our use of block-level dropout.**
>
>
> ### (3) Empirical Validation and Robustness
> Our design is justified by extensive empirical analysis, which **directly confronts the concern** that block dropout might lead to a *"severe degradation of prediction quality"* by affecting "critical blocks".
> *   *Toy Experiment:* As shown in **Figure 3**, our block-level dropout on a Gaussian mixture toy example produces results highly similar to using an explicit weak model. This visually confirms that, **rather than inducing unpredictable drift**, our method provides a stable guidance signal that effectively helps the process converge on the target distribution.
> *   *Robustness Analysis:*
>     *   *Qualitatively,* as shown in **Figure 9 (Appendix)** different random dropout strategies result in highly similar sampling trajectories, indicating **stable behavior**.
>     *   *Quantitatively,* our ablation studies reveal that even when a *single specific block is dropped at all timesteps*, the **performance degradation is minimal** for most blocks.
>
>     Crucially, **the independent selection of blocks to drop at each timestep** is precisely the mechanism that **prevents the very issue of quality degradation**. This stochastic process discourages the model from becoming overly reliant on any single "critical" block, thereby **mitigating the impact of continuously dropping any single block**. The empirical proof of this design's effectiveness is the **negligible output variance** detailed in our analysis in **W4** (see also **Appendix Table 7**), which confirms the **stability and reliability of this block-level dropout** as a guidance mechanism.
>
>
> *References:*
>
> *Avrahami et al. (2025): Stable flow: Vital layers for training-free image editing. CVPR 2025*.
>
> *Gal & Ghahramani (2016): Dropout as a Bayesian Approximation: Representing Model Uncertainty in Deep Learning. ICML 2016*.
>
> *Lou et al. (2024): Token caching for diffusion transformer acceleration. arXiv 2024*.
>
> *Huang et al. (2016): Deep Networks with Stochastic Depth. ECCV 2016*.
>
> *Karras et al. (2024): Guiding a diffusion model with a bad version of itself. NeurIPS 2024*.

---

> ### Author Response · Authors · 2025-11-23
> **Response to Reviewer 2Qir (2/4)**
>
> ## W2 (the second paragraph): Analysis of Block Dropout Impact and Visualization
>
> While our ablation study already provides a **(1) quantitative analysis** of the block dropout mechanism, we have now supplemented this with **(2) new qualitative visualizations of dropping each block** to directly address the concern about its perceptual impact.
>
> ### (1) Quantitative Analysis of the Dropout Mechanism
>
> To address the concern that the mechanism’s impact is *"under-analyzed,"* we highlight that we have already conducted a quantitative analysis of the impact of dropping specific blocks in our **Ablation Study (Section 4.5)**.
> *   **Block-wise Impact:** As shown in **Figure 7 (c, d)**, we evaluated the performance when dropping a *single, fixed block* throughout the entire denoising process. The results demonstrate that for most blocks, dropping any single one **does not lead to significant performance degradation**.
> *   **Independence and Robustness:** Notably, our $S^2$-Guidance applies block-dropping **independently at each timestep**. Unlike a fixed structural pruning, our stochastic approach ensures that no single block is consistently absent, thereby preventing error accumulation and **avoiding the risk of performance degradation**.
> *   **Variance Validation:** To provide concrete evidence of the method's stability and superiority, we conducted a run-to-run variance analysis and compared it against the CFG baseline on the HPSv2.1 benchmark.
>
>     | Method          | Mean(%)      | Var.       | Std. Dev | Coeff. of Var. |
>     | :-------------- | :-------- | :--------- | :------- | :------------- |
>     | CFG    | 30.48    | -        | -      | -            |
>     | **S$^2$-Guidance** | **30.86** | **0.000007** | **0.0026** | **0.84%**      |
>
>     The results show that our method not only **outperforms the baseline** but also achieves this with remarkable stability. The extremely low variance (on the order of $10^{-6}$) and a coefficient of variation under 1% confirm its **high reliability**.
>
> ### (2) Visualization of Dropping Each Block
>
> To intuitively demonstrate how these perturbations affect the final output, we provide a visual comparison on the SiT-XL (28 blocks) ImageNet 256x256 task. As shown in **Figure 15**, the image  generated with S²-Guidance exhibits **no perceivable degradation** compared to the baseline. Furthermore, the figure also shows that even when a **single, fixed block** is dropped for all timesteps—an extreme test case—the resulting image remains coherent and free of severe artifacts. This visual evidence robustly dispels the concern that block-level dropout introduces significant, disruptive perturbations.
>
> ---
>
> ## W3 (the third paragraph): On the Generality and Tuning-Free Nature of S$^2$-Guidance
> We would like to clarify a key point regarding the **tuning-free** and **plug-and-play** nature of our method. We highlight this via three points:
>
> ### (1) One-Time Validation, Not Per-Model Tuning
> The analysis of dropout probabilities was a one-time validation to establish a robust default (a drop ratio of approximately 10%). Once this default was identified, **no further parameter optimization was performed** when applying S$^2$-Guidance to new models. Our method is therefore **tuning-free at test time**, a significant practical advantage over approaches that require training auxiliary models or manually searching.
>
> ### (2) No Block Selection Required due to Stochasticity
> We would like to clarify that our final S$^2$-Guidance **does not require any manual block selection**. The method's robustness is instead derived from a simple principle: applying block-dropping *independently at each timestep*. This design ensures that the impact of any single dropped block is not sustained, **removing the need for model-specific tuning**. The stability of this fully automatic approach is empirically verified by our variance analysis (W2), which demonstrates consistently superior and reliable performance.
>
> ### (3) Empirical Evidence of Generality
> The strongest evidence for our claim of generality lies in the breadth of our evaluation. Using the **same unified configuration**, $S^2$-Guidance was successfully applied to **6 models**  across **3 tasks** (Class-conditional generation, T2I, T2V) and **4 benchmarks** (ImageNet, HPSv2.1, T2I-CompBench, VBench). The consistent superiority over baselines across this diverse set confirms that our method is indeed **a general, plug-and-play solution**.

---

> ### Author Response · Authors · 2025-11-23
> **Response to Reviewer 2Qir (3/4)**
>
> ## W4 (the fourth paragraph): Analysis of Stability, Consistency, and Motivation
>
> We address these points sequentially below, covering **(1) robustness to dropping key blocks**,  **(2) empirical consistency across runs**, and  **(3) motivation and advantages for our block-level dropout approach**.
>
> ### (1) Robustness to Dropping Key Blocks
>
> We would like to clarify that our final S$^2$-Guidance does not require any manual block selection. The method's robustness is instead derived from a simple but powerful principle: applying block-dropping **independently at each timestep**. Unlike a fixed structural pruning where a block is permanently removed, our stochastic approach ensures that no single block is consistently absent. This temporal independence prevents the accumulation of errors that might arise from dropping a critical block at a specific step, thereby **avoiding the risk of sustained performance degradation**. The stability is inherent to the process itself.
>
> ### (2) Empirical Variance Analysis
>
> To directly quantify the consistency of our method, we conducted a controlled experiment to isolate the variance caused by stochastic block dropping. We fixed the initial noise seed for a given prompt and performed multiple generation runs across the 3,200 images of the HPSv2.1 benchmark. The only source of variation was *the random mask sampled at each timestep*. As shown in the table below, **the run-to-run variance is vanishingly small**:
>
> | Method          | Mean(%)      | Var.       | Std. Dev | Coeff. of Var. |
> | :-------------- | :-------- | :--------- | :------- | :------------- |
> | CFG    | 30.48    | -        | -      | -            |
> | **S$^2$-Guidance** | **30.86** | **0.000007** | **0.0026** | **0.84%**      |
>
> The variance is on the order of $10^{-6}$, and the coefficient of variation is less than 1%. This empirically confirms that **the stochasticity introduced by S$^2$-Guidance does not destabilize the generation**. Despite the potential of dropping different blocks, the method converges to a highly consistent, high-quality mode.
>
> ### (3) Motivation and Advantages over Prior Work
>
> Our block-level dropout was developed to directly address **the limitations of AutoGuidance**, delivering a solution that is not only vastly **more practical in application** but also demonstrably **superior and more robust in performance**.
>
> * **Training-Free and "Plug-and-Play": Eliminating the Need for Auxiliary Models.**
>
>     This is the core motivation. AutoGuidance's method requires access to a separately trained weak model, but **acquiring such a specific, degraded version of a modern large-scale pretrained model is often infeasible**. This severely **limits the practical applicability and generality** of their method. In contrast, our approach is a **training-free mechanism** that can be **seamlessly applied** to any given pretrained model in a plug-and-play manner. We have validated this versatility across **6 different models** on **4 distinct benchmarks** in our extensive experiments.
>
> * **Superior Reliability and Effectiveness via Self-Guidance.**
>
>     As noted in AutoGuidance itself, **an improperly chosen weak model can fail to provide effective guidance**. Our approach circumvents this vulnerability by removing any dependency on external auxiliary models. Instead, it enables the model to guide itself **using a stochastic sub-network created on-the-fly**, generating guidance intrinsically from within the model itself. The resulting robustness is quantitatively validated by our analysis showing **negligible output variance** (**W4** or **Appendix Table 7**).
>
>
> ---
>
> ## W5 (the fifth paragraph): Comparison with PAG
>
> We have conducted a direct comparison which reinforces the advantages of our approach.
> Since PAG (Ahn et al., 2024) is specifically designed for U-Net architectures, we conducted a head-to-head comparison on a U-Net-based Stable Diffusion v1.5. Evaluated on HPSv2.1, our method demonstrates superior performance:
>
> | Method | HPSv2.1 (%) |
> | :--- | :--- |
> | CFG | 24.36 |
> | PAG | 24.65 |
> | **Ours** | **24.83** |
>
> These results show that  S$^2$-Guidance provides a more robust quality improvement.
>
> ---
>
> ## Minor
>
> Thank you for your meticulous feedback. We have revised the manuscript accordingly to make our phrasing more precise.

---

> ### Author Response · Authors · 2025-11-23
> **Response to Reviewer 2Qir (4/4)**
>
> ## Q1: On the Formulation Difference from Standard CFG
> Thank you for this insightful question. The formulation difference you've identified is, in fact, the core of our method. We designed this subtractive form to **introduce a critical correction** that overcomes the inherent limitations of standard Classifier-Free Guidance (CFG) and **directly refines its suboptimal results**. Our rationale is built on the following points:
>
> * **The Known Limitations of Standard CFG's Linear Trajectory**
> CFG's formulation restricts the guidance to a **linear interpolation/extrapolation** between the conditional and unconditional model predictions. This approach has two well-documented issues:
>     *   **Recent Literature:** As highlighted in recent works (*Jin et al., 2025; Chung et al., 2024*), simple linear extrapolation often overshoots the true data manifold, leading to common failure modes (e.g., visual artifacts).
>     *   **Empirical Observation:** We empirically demonstrate this exact issue in our toy example (**Figure 3**), which visually confirms that CFG can produce suboptimal results by straying from the ideal data distribution.
>
> * **Our Solution: Corrective Guidance via an Implicit Weak Model**
>
>     To address this, our work aligns with advanced methods that aim to correct the guidance geometry. We draw inspiration from the principle of **weak-model** (*Karras et al., 2024*). The principle behind this approach is that a weak model is expected to make similar errors as the main model in low-probability, challenging regions. By identifying the direction of these errors, one can actively steer the main model **away** from them.
>
>     Our key insight is that **the subtracted term** in our formulation originates from **an implicit weak model**, which we generate dynamically via stochastic block dropping. **The prediction from this weak model serves to identify potential failure modes**—the low-density, suboptimal regions where the full model is prone to error. Therefore, the **act of subtraction** is crucial: it applies a repulsive force against these failure modes, actively **steering the sampling trajectory away from bad regions** and thereby **directly correcting the suboptimal linear path of standard CFG.**
>
> In essence, this transforms the rigid linear path of CFG into a dynamic, non-linear trajectory that steers the generation back towards the high-density data manifold, leading to demonstrably higher-quality and more faithful results.
>
>
> *References:*
>
> *Jin et al. (2025): Angle domain guidance: Latent diffusion requires rotation rather than extrapolation. ICML 2025*.
>
> *Chung et al. (2024): Cfg++: Manifold-constrained classifier free guidance for diffusion models. ICLR 2025*.
>
> *Karras et al. (2024): Guiding a diffusion model with a bad version of itself. NeurIPS 2024*.
>
> ---
>
> ## Q2: Rationale and Applicability to Other Methods
>
> We explain our rationale by linking the concept of **Weak Supervision** with the empirical observation of **Error Correlation**.
>
> **Rationale: Weak Models as Corrective Signals**
> The core rationale for our subtractive formulation is to utilize the weak sub-network as a supervisory signal to correct the strong model's tendency to drift into low-quality regions. This approach is grounded in the broader principle that weak models can effectively guide strong models (*Burns et al. 2023, Karras et al., 2024*). In our framework, subtracting the sub-network prediction applies a **corrective force** that exposes and neutralizes the model's uncertainty, steering the sampling trajectory back towards high-fidelity modes.
>
> **Why Sub-networks Work: Capturing Correlated Errors**
> This subtraction is effective because of **Error Correlation**. A useful weak model is expected to share similar, yet more pronounced, failure modes with its stronger counterpart. Our Gaussian Mixture Toy Experiments (Figure 3) demonstrate that stochastic sub-networks function precisely as such weak models. They exhibit failure modes (e.g., mode collapse, drift) that are directionally correlated with the suboptimal results of CFG. Therefore, subtracting $D_{\text{sub}}$ effectively cancels out these specific errors, rectifying predictions without needing an external reference model.
>
>
> **Applicability to Other Bad-Version Methods**
> We experimentally applied a similar ω-scaled subtraction to PAG and observed no significant performance improvement. However, the validity of our underlying principle is demonstrated by AutoGuidance (*Karras et al., 2024*), which has proven to be effective.
>
> *References:*
>
> *Burns et al. (2023): Weak-to-strong generalization: Eliciting strong capabilities with weak supervision. arXiv 2023*.
>
> *Karras et al. (2024): Guiding a diffusion model with a bad version of itself. NeurIPS 2024*.
>
> ---
>
> Please let us know if any concerns remain unaddressed; we are happy to discuss them.

---

### Official Review · Reviewer_amct · 2025-10-31

**Soundness:** 2
**Presentation:** 3
**Contribution:** 3
**Rating:** 6
**Confidence:** 3

**Summary:**

This paper introduces S²-Guidance, a training-free modification to classifier-free guidance (CFG) for diffusion models. At each denoising step, the method forms a stochastic “weak” sub-network by randomly dropping some transformer blocks and subtracts its prediction (scaled by ω) from the standard CFG result. The goal is to mitigate CFG’s over-confident or semantically inconsistent generations while preserving distributional fidelity. The paper reports consistent improvements on several text-to-image and text-to-video benchmarks with transformer-based backbones.

**Strengths:**

- Training-free inference procedure that does not require auxiliary models.
- Broad experimental coverage across multiple transformer-based diffusion backbones (images and video) with clear qualitative examples.
- Ablation studies on several design choices (ω, drop ratio) and comparisons against a range of CFG variants.
- The idea is easy to understand and implement for transformer architectures, and the empirical results are consistently positive.

**Weaknesses:**

- Computational overhead: per denoising step the method requires an additional denoiser call (three evaluations vs. two for standard CFG). The efficiency claim is mainly relative to a heavier “naive” multi-sample variant rather than to CFG. The phrasing of the term "efficient" in the main text is therefore slightly misleading.
- Architecture dependence: all demonstrations are on transformer-based diffusion models and the mechanism relies on dropping entire residual blocks with fixed shapes. It is unclear how to apply the same stochastic sub-network idea to UNet/CNN backbones without nontrivial changes.
- Theoretical framing is largely interpretive. The Bayesian idea discussed in the appendix is suggestive but not shown to tighten score estimation or likelihood in a formal sense.
- Scope of validation: no compute/runtime comparisons versus CFG are provided, and the variance introduced by the stochastic dropping is not discussed.

**Questions:**

1. Applicability to UNet/CNN architectures: please explain concretely how the stochastic sub-network would be constructed for UNet-style diffusion models (e.g., which modules are droppable without breaking shapes/skip connections) and provide some experiments in this direction to support the generality claim.
2. Compute efficiency: please report runtime or FLOP ratios relative to standard CFG for a typical sampling trajectory, and include wall-clock time where possible.
3. Theory: in the Bayesian derivation, why should “repulsion from the posterior mean” improve sample fidelity or score accuracy?
5. Output variance: does the stochastic mask introduce noticeable variance across runs with identical seeds?

---

> ### Author Response · Authors · 2025-11-23
> **Response to Reviewer amct (1/4)**
>
> We sincerely thank you for your constructive comments and for your recognition of our work's strengths, including its *training-free design*, *broad experimental coverage*, and *consistently positive results*. Below, we provide a point-by-point response to the weaknesses and questions you raised.
>
> ---
>
> ## W1: Efficiency Claim and Computational Cost
>
> Thank you for the valuable suggestion to clarify our use of the term "efficient." In the revised manuscript, we have clarified that our discussion of "efficiency" is primarily in comparison to the naive multi-sample variant, thereby avoiding any potential misunderstanding relative to standard CFG.
>
> Regarding the computational cost relative to CFG, we would like to clarify that our design ensures **this additional denoiser call is justified by a more effective use of computational resources**, a claim supported by the following analyses:
>
> *   **Superior Performance at an Equivalent Computational Budget**
>
>     As shown in our **Table 4 (Appendix)**, when the total number of forward passes is kept constant, our method consistently achieves higher scores on both human preference (HPSv2.1) and aesthetic quality (QAlign). This demonstrates that for a fixed computational budget, our method yields a superior result.
>
> *   **Additional Systematic Trade-off Analysis**
>
>     To further address this concern, we have added an additional performance-vs-efficiency analysis in **Figure 14 (Appendix)**. This plot illustrates that our method establishes a **more favorable performance-efficiency frontier**. S$^2$-Guidance consistently occupies the **top-left region**, indicating it achieves a higher level of performance for any given level of computational expenditure compared to the alternatives.
>
> ---
>
> ## W2: Architecture Dependence
> First, we would like to clarify that our primary focus on **DiT architectures** was a deliberate choice. As these backbones represent the current **state-of-the-art in generative modeling** (e.g., Flux, Wan), our goal was to validate our method's effectiveness first and foremost on these frontier models.
>
> Regarding the **application of our method to other architectures**:
>
> *   For **CNN**-based backbones, these architectures fall outside the primary scope of our work. As our title suggests, the scope of our investigation is guidance techniques specifically for diffusion models.
>
> *   For **UNet** architectures, our method **can be applied directly** and **without nontrivial changes**. For example, the classic Stable Diffusion v1.5 model, while UNet-based, naturally incorporates transformer blocks within its architecture. We can therefore directly apply the same stochastic sub-network mechanism by dropping these internal transformer blocks.
>
>     The results from our new experiment on SD v1.5 show **a clear and consistent improvement**:
>
>     | Method          | HPSv2.1(%) $\uparrow$ |
>     | :-------------- | :------------------------- |
>     | CFG    | 24.35                      |
>     | **Ours** | **24.83**                  |
>
> This new result confirms that S$^2$-Guidance is not limited to DiT architectures. Its **seamless applicability to UNet-based models** demonstrates the **robustness and generalizability** of our approach.

---

> > ### Comment · Reviewer_amct · 2025-11-24
> > **Overclaiming concerns**
> >
> > Regarding your statement:
> >
> > > “This new result confirms that S-Guidance is not limited to DiT architectures. Its seamless applicability to UNet-based models demonstrates the robustness and generalizability of our approach.”
> >
> > The SD-1.5 experiment only shows that your method can drop the transformer layers inside a hybrid UNet model. This is not the same as demonstrating applicability to UNet or CNN architectures in general. Many diffusion models use purely convolutional UNets without any transformer blocks, and your experiment does not show how S-Guidance would apply to such architectures. The improvement you report is also small, which is expected given that a lot of SD-1.5’s parameters reside in the CNN backbone rather than in the attention modules that you drop. It seems that you are using terms diffusion and transformers as if one implies the other, which is not the case.
> >
> > As a result, the paper currently overclaims generality, as I have originally stated. What your experiments clearly support is that your method works for transformer-based diffusion models and hybrid models that already contain transformer layers. This is fine, but I would strongly encourage the authors to adjust the wording in the paper to reflect this, rather than implying universal applicability to all UNet or CNN diffusion backbones.

---

> > > ### Author Response · Authors · 2025-11-24
> > > **Response to the Follow-up Comment on Architecture Dependence**
> > >
> > > We sincerely thank you for the prompt and insightful follow-up. We fully accept your valuable observation regarding the distinction between "general diffusion models" and "transformer-based/hybrid diffusion models." To ensure our claims are precise, we have revised the manuscript to **explicitly scope our method to "Transformer-based/Hybrid Diffusion Models**."
> > >
> > > While we have refined the claimed scope, we respectfully highlight that **this scope encompasses the dominant paradigm in high-fidelity diffusion generation**. The field has shifted towards **DiT** architectures (e.g., SD3, SD3.5, Flux, HunyuanVideo, Wan2.2, Qwen-Image, Sora2) due to their scalability. Therefore, while the scope is technically narrower than "all diffusion models," it covers the most critical frontiers of generative modeling research.
> > >
> > > We are deeply grateful for your time and continued engagement in this discussion. Your feedback has significantly improved the paper's precision and rigor. We have strictly adjusted our wording to reflect this specific scope, ensuring the paper's claims are fully aligned with the experimental evidence.
> > > If you have any further concerns, we would be more than happy to address them.

---

> ### Author Response · Authors · 2025-11-23
> **Response to Reviewer amct (2/4)**
>
> ## W3: Clarification on Score Estimation Tightening
>
> We sincerely appreciate your detailed feedback and your rigorous scrutiny regarding the theoretical foundation.
> We acknowledge that our Bayesian derivation serves primarily as an interpretive framework.
>
> However, we emphasize that **providing a step-by-step formal proof of likelihood tightening** is a **known open challenge in generative modeling** (*Song et al., 2020b*,*Theis et al., 2016*).
> Even **foundational works** in this field rely on **alternative validation methods** due to this limitation: **Classifier-Free Guidance (CFG)** (*Ho & Salimans, 2022*) primarily established its validity through **rigorous empirical observation**, while **AutoGuidance** (*Karras et al., 2024*) relied heavily on **toy examples** to demonstrate its mechanism.
>
> Following this established rigorous standard, we substantiate our claim that $S^2$-Guidance "tightens" the estimation by integrating **both** analysis strategies—*(1) precise toy verification* and *(2) broad empirical testing*—augmented by *(3) a rigorous interpretive framework*. Our validation approach covers:
>
> **(1) Closed-form Verification on Toy Examples**
> Since the exact score is intractable for real images, we utilize toy examples with **closed-form solutions** (1D/2D Gaussian Mixtures) as the "ground truth" for theoretical verification.
> * *Evidence:* As shown in **Figure 3** and **Figure 9**, standard estimators produce "loose" distributions with drifted modes. $S^2$-Guidance, by repelling from the sub-network prediction, successfully rectifies these trajectories to align precisely with the ground truth modes. This provides concrete proof that our mechanism mathematically **tightens the score estimation towards the true mode in controlled environments**.
>
> **(2) Extensive Empirical Validation**
> We verify that these properties translate to high-dimensional space through extensive experimentation across **6 models** (SiT, SD3, SD3.5, Flux, Wan-1.3B, Wan-14B), **3 tasks** (Class-conditional, T2I, T2V), and **4 benchmarks** (HPSv2.1,T2I-ComBench, VBench, Imagenet).
> * *Consistency:* The consistent superiority in fidelity metrics (HPSv2.1, VBench, Inception Score) confirms that **the "tightening" effect is a fundamental correction of the generative process**, not an artifact of a specific setup.
>
> **(3) Interpretive Framework: Repulsion from the "Center of Uncertainty"**
> Finally, we ground these results in our Bayesian derivation, leveraging the established principle of **"Dropout as a Bayesian Approximation"** (*Gal & Ghahramani, 2016*). Within this framework, the stochastic sub-network prediction serves as a sample from the approximate posterior distribution.
> * *The "Loose" Component:* As discussed in our Appendix, the posterior mean corresponds to a "center of mass" of the model's belief. It represents a "safe" but often "low-quality output"  that lies in regions of high epistemic uncertainty.
> * *The Tightening Mechanism:* By subtracting this component, our method effectively filters out this uncertainty. This mathematically tightens the score estimation by removing the tendency to settle in "suboptimal regions" and forcing the trajectory to commit to "high-quality regions" on the data manifold.
>
> *References:*
>
> *Song et al. (2020b): Score-Based Generative Modeling through Stochastic Differential Equations. ICLR 2021*.
>
> *Theis et al. (2016): A note on the evaluation of generative models. ICLR 2016*.
>
> *Ho & Salimans (2022): Classifier-Free Diffusion Guidance. NeurIPS 2022*.
>
> *Karras et al. (2024: Guiding a diffusion model with a bad version of itself. NeurIPS 2024*.
>
> *Gal & Ghahramani (2016): Dropout as a Bayesian Approximation. ICML 2016*.

---

> ### Author Response · Authors · 2025-11-23
> **Response to Reviewer amct (3/4)**
>
> ## W4: Scope of Validation
>
> To broaden the scope of validation, we have conducted two additional analyses addressing the practical aspects of our method: *(1) a comparison of computational cost* and *(2) an analysis of the variance from stochastic dropping*.
>
> **(1) Computational Cost Comparison**
>
> We measured the runtime and FLOPs of our method against standard CFG for a 28-step inference task.
>
> | Method | Runtime (End-to-End) |  FLOPs (Transformer) |
> | :----- | :--------------------------- | :---------------- |
> | CFG    | 29.2 s                       | 168.4 TFLOPs      |
> | Ours   | 40.2 s                       | 237.6 TFLOPs      |
>
> The results show an approximate 40% overhead in both runtime and FLOPs.
> While this entails a degree of additional overhead, we re-emphasize that it is justified by a **superior performance-efficiency trade-off**, as demonstrated in our response to **W1**. Our method achieves significantly **higher quality for an equivalent computational budget**, establishing it as a **more advanced and practical choice** for maximizing generation quality.
>
> **(2) Analysis of Variance from Stochastic Dropping**
>
> We conducted a dedicated experiment to precisely quantify this effect. Our findings show that **the variance introduced is extremely low**.
>
> *Experimental Setup:*
> To isolate the variance from stochastic dropping, we fixed the initial noise seed for a given prompt across multiple generation runs. Further details on this setup are provided in **Appendix**.
>
> *Quantitative Results:*
> Our analysis reveals that the run-to-run variance is negligible. The quantitative results are summarized below:
>
> | Method          | Mean(%)      | Var.       | Std. Dev | Coeff. of Var. |
> | :-------------- | :-------- | :--------- | :------- | :------------- |
> | CFG    | 30.48    | -        | -      | -            |
> | **S$^2$-Guidance** | **30.86** | **0.000007** | **0.0026** | **0.84%**      |
>
> As shown, the variance is on the order of $10^{-6}$, and the coefficient of variation is less than 1%. This indicates an extremely high degree of stability and output consistency.
>
> ---
>
> ## Q1: Construction and Generality for UNet Architectures
>
> Our method is **naturally applicable to UNet-style diffusion models**, as they are hybrid architectures that incorporate transformer blocks. To answer your specific question on construction: the key is that these internal transformer blocks are shape-preserving, meaning they process a feature map and return a new one of the exact same dimensions. The UNet's skip connections operate on the outputs of these modules. Therefore, by dropping a subset of these transformer blocks, we can construct the stochastic sub-network **without breaking shapes or interfering with the skip connection** pathways.
>
> To validate this approach and directly **support our generality claim**, we conducted a new experiment on SD v1.5, which (as detailed in our response to **W2**) shows a clear and consistent performance improvement over the baseline. This successful application to a UNet-based model, achieved without any non-trivial changes, provides strong empirical evidence for the **generalizability and robustness** of our proposed method.
>
> ---
>
> ## Q2: Compute Efficiency
> We have provided a detailed analysis of runtime and FLOPs in our response to **W4 (1)**. For your convenience, we reproduce the key results and our analysis here.
> | Method | Wall-clock Time |  FLOPs (Transformer) |
> | :----- | :--------------------------- | :---------------- |
> | CFG    | 29.2 s                       | 168.4 TFLOPs      |
> | Ours   | 40.2 s                       | 237.6 TFLOPs      |
>
> The results show an approximate 40% overhead in both runtime and FLOPs.
> While this entails a degree of additional overhead, we re-emphasize that it is justified by a **superior performance-efficiency trade-off**, as demonstrated in our response to **W1**. Our method achieves significantly **higher quality for an equivalent computational budget**, establishing it as a **more advanced and practical choice** for maximizing generation quality.

---

> ### Author Response · Authors · 2025-11-23
> **Response to Reviewer amct (4/4)**
>
> ## Q3: Mechanism Explanation on How "Repulsion from Posterior Mean" Improves Fidelity
>
> In traditional regression tasks, the posterior mean is indeed the optimal estimator for minimizing Mean Squared Error (MSE). However, we wish to clarify that in the context of generative modeling, approximating the posterior mean often leads to low fidelity results, whereas "repelling from the mean" is key to improving sample quality.
> Here is the mechanism explanation based on Bayesian derivation and existing literature:
>
> **(1) Posterior Mean Corresponds to "Conservative Fit" and "Safe Output"**
> In our Bayesian framework, the sub-network ensemble's posterior mean $\mu_{post}$ captures the model's uncertainty.
>
> * **Conservative Fit:** As *Karras et al. (2024)* point out, maximum likelihood training tends to produce a **"conservative fit"**, where the model attempts to cover all training samples.
> * **Center of Uncertainty:** In our derivation, $\mu_{post}$ represents this "safe" but "low-quality output". It lies at the "center of uncertainty" of the probability density, visually manifesting as a blurred or averaged result lacking high-frequency details.
>
> **(2) Repulsion Mechanism Implements "Sharpening" and "Lowering Temperature"**
> Therefore, "repulsion from the posterior mean" is not intended to improve Score Accuracy with respect to the original distribution, but to correct the aforementioned conservativeness.
>
> * **Sharpening:** *Bradley & Nakkiran (2024)* prove that CFG, by *"raising ... to a power greater than one"*, is essentially performing a "sharpening" operation.
> * **Avoidance of Uncertainty:** Our method employs the following correction formula:
>     $$
>     \tilde{D} = D - \omega \cdot \mu_{post}
>     $$
>     Mathematically, this uses $\mu_{post}$ to locate and **"avoid regions of high model uncertainty"**.
> * **Temperature Reduction:** This effectively executes the **"lowering the sampling temperature"** described by *Karras et al. (2024)*, trading diversity (Likelihood) for clearer modes.
>
> **Summary:**
> "Repulsion from the posterior mean" improves fidelity because, in generative tasks, the posterior mean represents the "conservative fit" component that hinders high-fidelity generation. Our method effectively "sharpens" and deblurs the sampling distribution by repelling this component, thereby achieving higher perceptual quality.
>
> *References:*
>
> *Bradley & Nakkiran (2024): Classifier-free guidance is a predictor-corrector. arXiv 2024*.
>
> *Karras et al. (2024): Guiding a diffusion model with a bad version of itself. NeurIPS 2024*.
>
> ---
>
> ## Q4: Output Variance
>
> The stochastic mask **does not introduce noticeable variance** across runs with identical seeds. We have provided a detailed **quantitative analysis** in our response to **W4 (2)**, which confirms **the variance is numerically negligible**.
>
> To more directly address the question of *noticeable* variance, we highlight our **qualitative toy experiment**. In the 2D Gaussian mixture model (**Appendix, Figure 10**), repeated runs initiated from the same seed produce **visually indistinguishable distributions**. This provides clear, intuitive evidence that while the mask sequence is random, the generation process consistently converges to the same stable output mode.
>
> ---
>
> Please let us know if any concerns remain unaddressed; we are happy to discuss them.

---

### Official Review · Reviewer_dNUy · 2025-10-31

**Soundness:** 3
**Presentation:** 3
**Contribution:** 2
**Rating:** 4
**Confidence:** 4

**Summary:**

The paper introduces $S^2$-Guidance (Stochastic Self-Guidance), a training-free guidance algorithm for diffusion models. The authors have empirically shown that Classifier-Free Guidance (CFG) can produce suboptimal results. To address this issue, they proposed activating subnetworks of the same model through stochastic block dropping during denoising to obtain a “weak” prediction that corrects the CFG direction. The proposed $S^2$-Guidance improves fidelity across class-dependent ImageNet, text-to-image, and text-to-video generation, outperforming CFG and current guidance algorithms.

**Strengths:**

- The paper is well-written and easy to understand.
-  Activating an internal “weak” predictor by randomly omitting blocks during inference is simple but effective. Unlike distillation-based methods and external weak models, this algorithm is training-free and plug-and-play. It can therefore be applied directly to the generation process, which is currently based on CFG.
- An additional stochastic forward pass per step keeps complexity low compared to many other alternatives with guidance or self-ensemble, making the ratio between quality and computational effort efficient in practice.

**Weaknesses:**

- In Appendix A, the authors provides an analysis of $S^2$-Guidance and Naive $S^2$-Guidance. The authros "posit" that $S^2$-Guidance is an approximately unbiased estimator of the "expected guidance" $G_{Naive}$. However, the "unbiasedness" of $G_{S^2-Guidance}$ is assumed, not shown. In addition,
$\mu_{post}$ denotes a Bayesian posterior mean, but the algorithm actually only defines an empirical average of the predictions of the sub-network obtained by randomly dropping modules, which cannot follw eq. (25).
- The author must argue that the variance of $G_{S^2-Guidance}$ is small, or specify concentration bounds so that one guidance per step is reliable.
- A randomly drawn sub-network leads to a deviation in the trajectory. Sensitivity and stability under randomness have not yet been sufficiently investigated in the paper.
- For videos, stochastic guidance can lead to temporal flickering or shifts in object identity. Therefore, please add temporal stability metrics to verify the quality of the generated video.
- The proposed $S^2$-Guidance requires an extra forward pass per step. Thus, FLOPs and peak memory requirements must be reported.

**Questions:**

- How large in the variance of the single stochastic block-dropping operation?
- How dose $S^2$-Guidance affect long composition prompts, text rendering, or high-frequency textures?

---

> ### Author Response · Authors · 2025-11-23
> **Response to Reviewer dNUy (1/4)**
>
> We sincerely thank you for your constructive comments and for your recognition of our work's strengths, including the *well-written presentation*, the *simple and effective algorithm*, and the *efficient ratio between quality and computational effort*. Below, we provide a point-by-point response to the weaknesses and questions you raised.
>
> ---
>
> ## W1: Clarification on Unbiasedness and Posterior Mean Estimation
>
> Thank you for the rigorous scrutiny. We address these theoretical points by **(1) formally proving unbiasedness** and **(2) clarifying our posterior mean estimation**.
>
> ### (1) Formal Proof of Unbiasedness
>
> We acknowledge that the term "posit" may have unintentionally obscured the formal nature of our derivation, and we have revised the manuscript to ensure precision. We explicitly clarify that we do not assume unbiasedness; rather, **we formally derive that both $G\_{S^2}$ and $G\_{\text{Naive}}$ are unbiased Monte Carlo estimators of the same theoretical population mean.**
>
> Let $\theta$ be the model parameters and $p(m)$ be the distribution of binary masks. Following *Gal & Ghahramani (2016)*, the stochastic block-dropping process induces a variational distribution $q(\tilde{\theta})$ over the parameter space, where $\tilde{\theta} = \theta \odot m$.
>
> We define the **Theoretical Expected Guidance** (the population mean) as the exact predictive mean under this induced distribution:
>
> $$
> \mathcal{G}^* \triangleq \omega \cdot \mathbb{E}\_{q(\tilde{\theta})}[D(x\_t|c; \tilde{\theta})] \equiv \omega \cdot \mathbb{E}\_{m \sim p(m)}[\hat{D}\_{\theta}(x\_t|c, m)]
> $$
>
> Using the linearity of expectation, we derive the properties of our two estimators:
>
> * Naive $S^2$-Guidance ($G\_{\text{Naive}}$), which averages $N$ i.i.d. samples:
> $$
> \mathbb{E}[G\_{\text{Naive}}] = \mathbb{E}\left[ \frac{\omega}{N} \sum\_{i=1}^N \hat{D}(\cdot, m\_i) \right] = \frac{\omega}{N} \sum\_{i=1}^N \mathbb{E}\_{p(m)}[\hat{D}(\cdot, m)] = \mathcal{G}^*
> $$
>
> * $S^2$-Guidance ($G\_{S^2}$), which uses a single sample ($N=1$):
> $$
> \mathbb{E}[G\_{S^2}] = \mathbb{E}[\omega \cdot \hat{D}(\cdot, m\_t)] = \omega \cdot \mathbb{E}\_{p(m)}[\hat{D}(\cdot, m)] = \mathcal{G}^*
> $$
>
> **Conclusion:** Since $\mathbb{E}[G\_{S^2}] = \mathbb{E}[G\_{\text{Naive}}] = \mathcal{G}^*$,
>
> both methods are mathematically **unbiased Monte Carlo estimators** of the target $\mathcal{G}^*$, differing only in variance.
>
> ### (2) Clarification on Posterior Mean Estimation
>
> We clarify the theoretical alignment between the posterior mean $\mu\_{post}$ and the empirical average calculated by our algorithm. These are intrinsically connected through the methodology used to resolve intractable integrals in deep learning.
>
> **Theoretical Connection:**
> We rely on the Variational Inference framework established by *Gal & Ghahramani (2016)*. The stochastic block-dropping process induces a variational distribution $q(\tilde{\theta})$ over the parameter space (structurally similar to *Stochastic Depth (Huang et al., 2016)*). Theoretically, the posterior predictive mean is the integral over this induced distribution:
>
> $$
> \mu\_{post} = \int D(x\_t|c; \tilde{\theta}) q(\tilde{\theta}) d\tilde{\theta}
> $$
>
> Since this integral is analytically intractable for deep neural networks, we must rely on numerical approximation methods established in the literature.
>
> **Literature Support:**
> Computing such high-dimensional integrals via empirical averaging over diverse predictive hypotheses is a standard practice in deep learning. This paradigm is supported by extensive literature, including explicit methods like *Deep Ensembles* (*Lakshminarayanan et al., 2017*) and efficient implicit ensembles like *MC-Dropout* (*Gal & Ghahramani, 2016*), *Snapshot Ensembles* (*Huang et al., 2017*), and *BatchEnsemble* (*Wen et al., 2020*). These works collectively establish that averaging over stochastic sub-states or ensemble members effectively approximates the predictive posterior. **Our algorithm is a direct application of this principle**, leveraging the variational ensemble induced by stochastic block-dropping to estimate the integral.
>
> We have revised Appendix A to explicitly state that **our empirical average serves as the Monte Carlo estimator for the intractable posterior integral**, supported by these references.
>
> *References:*
>
> *Gal & Ghahramani (2016): Dropout as a Bayesian Approximation. ICML 2016.*
>
> *Lakshminarayanan et al. (2017): Simple and Scalable Predictive Uncertainty Estimation using Deep Ensembles. NeurIPS 2017.*
>
> *Huang et al. (2017): Snapshot Ensembles: Train 1, get M for free. ICLR 2017.*
>
> *Wen et al. (2020): BatchEnsemble: An Alternative Approach to Efficient Ensemble and Lifelong Learning. ICLR 2020.*
>
> *Huang et al. (2016): Deep Networks with Stochastic Depth. ECCV 2016.*

---

> > ### Author Response · Authors · 2025-11-23
> > **Response to Reviewer dNUy (2/4)**
> >
> > ## W2 & Q1: Reliability of Single-Step Guidance and Theoretical Justification
> >
> > We demonstrate that a single guidance step is reliable because the variance is empirically negligible and theoretically bounded by the architecture's inherent redundancy. We address this by **(1) explaining the structural overlap**, **(2) formally deriving the variance bound**, and **(3) providing empirical verification**.
> >
> > ### (1) Theoretical: Stability Derived from Structural Overlap
> >
> > Our method constructs a **stochastic sub-network** by dynamically skipping a subset of blocks. The reliability of a single stochastic pass is grounded in **High Structural Overlap**. Since the drop ratio is low (~10%), any two random sub-networks share the vast majority (approx. 90%) of their active parameters and computational paths. Consequently, they are highly correlated variants of the full model rather than independent estimators.
> >
> > ### (2) Formal Derivation: Structural Concentration Bound
> >
> > We formally prove this stability using the **Efron-Stein inequality**, applying it to the **empirical feature redundancy** observed in DiT literature (e.g., *Chen et al., 2024*).
> >
> > * **Setup:** Let $m = (b_1, \dots, b_L)$ be the random mask vector, where $b_l \in \{0, 1\}$ indicates whether the $l$-th block is active. Let $\hat{D}(m)$ be the sub-network output.
> > * **Bound:** The Efron-Stein inequality bounds the variance of $\hat{D}(m)$:
> >     $$
> >     \text{Var}(\hat{D}(m)) \le \frac{1}{2} \sum_{l=1}^L \mathbb{E}[\|\hat{D}(m) - \hat{D}(m^{(l)})\|^2]
> >     $$
> >     where $m^{(l)}$ represents the mask $m$ with the $l$-th bit toggled (flipped). The term $\|\hat{D}(m) - \hat{D}(m^{(l)})\|$ measures the **functional difference** caused by toggling the state of a single block $l$.
> > * **Redundancy Premise:** Literature confirms that deep DiT blocks exhibit high **feature redundancy**, meaning the contribution of a single block to the global semantic representation is marginal. Mathematically, this implies the functional difference is bounded by a small constant $\epsilon$: $\|\hat{D}(m) - \hat{D}(m^{(l)})\| \le \epsilon$.
> >
> > **Conclusion:** Substituting this bound, we obtain:
> > $$
> > \text{Var}(\hat{D}(m)) \le \frac{1}{2} \sum_{l=1}^L \epsilon^2 = \frac{L}{2} \epsilon^2
> > $$
> > This demonstrates that the total variance is tightly constrained by the sum of these small squared perturbations, theoretically guaranteeing that a single stochastic pass remains close to the expectation.
> >
> > ### (3) Empirical: Quantitative Negligible Variance Analysis
> >
> > To strictly verify this reliability, we fixed the initial noise seed and generated images for the HPS v2.1 benchmark (3,200 prompts) multiple times, varying *only* the stochastic masks. The results confirm our theoretical derivation:
> >
> > | Method | Mean (%) | Var.  | Std. Dev | Coeff. of Var.  |
> > | :--- | :--- | :--- | :--- | :--- |
> > | CFG    | 30.48    | -        | -      | -            |
> > | **Ours** | **30.86** | **0.000007** | **0.0026** | **0.84%** |
> >
> > With a coefficient of variation below **1%**, the stochasticity is empirically negligible, confirming that a single guidance step is highly reliable and stable.
> >
> > ---
> >
> > ## W3: Sensitivity and Stability under Randomness
> >
> > We address the concern regarding sensitivity to randomness with both quantitative reference and qualitative visualizations.
> >
> > ### (1) Quantitative Stability
> >
> > As demonstrated in the table provided in our response to **W2 & Q1**, the run-to-run variance of our method is on the order of $10^{-6}$. This confirms that the model is insensitive to the specific random mask sampled at each step and consistently converges to the same high-quality mode.
> >
> > ### (2) Qualitative: Trajectory Consistency
> >
> > We refer the reviewer to **Figure 10** in the Appendix. It visualizes the final distribution of samples generated with Naïve $S^2$-Guidance (averaged multiple sub-networks) versus our single-step $S^2$-Guidance. When repeating the generation from a fixed initial noise seed, the resulting sample distributions are **visually indistinguishable** from one another. This provides strong intuitive evidence that the method consistently finds the same stable convergence point, regardless of the random mask sequence.
> >
> > In conclusion, our method demonstrates **exceptional stability and low sensitivity** to its inherent randomness. This is substantiated by both negligible quantitative variance and visually identical outputs in repeated runs, confirming the **robustness and reliability** of the guidance mechanism.

---

> > > ### Author Response · Authors · 2025-11-23
> > > **Response to Reviewer dNUy (3/4)**
> > >
> > > ## W4: Temporal Stability Metrics
> > > We are pleased to report that, far from introducing instability, our method demonstrably improves temporal quality compared to the standard CFG baseline.
> > >
> > > | VBench Metric             | CFG    | Ours         |
> > > | :------------------------ | :----- | :----------- |
> > > | Subject Consistency (↑)   | 0.9653 | **0.9657**   |
> > > | Temporal Flickering (↑)   | 0.9949 | **0.9952**   |
> > > | Background Consistency (↑)| 0.9546 | **0.9580**   |
> > > | Dynamic Degree (↑)        | 0.7361 | **0.7917**   |
> > >
> > > As the table clearly demonstrates:
> > >
> > > *   Our method achieves higher scores in **Subject Consistency** and **Temporal Flickering**, confirming that S²-Guidance effectively mitigates identity shifts and reduces flickering artifacts.
> > > *   Beyond these core temporal metrics, we also report on **Dynamic Degree** to reflect overall video quality. Our method not only improves **Background Consistency** but also achieves a massive **+7.6%** boost in this crucial metric.
> > >
> > > In summary, these results provide strong empirical evidence that our method enhances not only key **temporal metrics** but also the  **dynamism** of the generated videos.
> > >
> > > ---
> > >
> > > ## W5: Computational Cost and Peak Memory
> > >
> > > We have conducted a direct comparison of FLOPs, runtime, and peak memory requirements, as requested.
> > >
> > > * **Computational Cost & Memory Comparison**
> > >
> > >     To provide a clear picture, we benchmarked standard CFG against our S²-Guidance on a T2I task with 28 inference steps. The results are summarized below:
> > >
> > >     | Method | Total Runtime (End-to-End) | Transformer FLOPs | Peak GPU Memory (Allocated) |
> > >     | :----- | :--------------------------- | :---------------- | :-------------------------- |
> > >     | CFG    | 29.2 s                       | 168.4 TFLOPs      | ~33.8 GB                    |
> > >     | Ours   | 40.2 s                       | 237.6 TFLOPs      | ~33.8 GB                    |
> > >
> > >     As the data shows, our method incurs an overhead of approximately 40% in both runtime and computational FLOPs for the same number of sampling steps.
> > >
> > >     Regarding peak memory, our method **does not increase the requirement**. This is because the two forward passes (the full model and the sub-network) are executed **sequentially** within each denoising step. The memory from the first pass is released before the second begins, meaning the peak memory footprint remains identical to that of a single standard CFG evaluation.
> > >
> > > * **Performance-Efficiency Trade-off**
> > >
> > >     While this entails a degree of additional overhead, we emphasize that it is justified by a **superior performance-efficiency trade-off**. To substantiate this, we have added a new performance-vs-efficiency analysis in Figure 14 (Appendix). This plot illustrates that our method establishes a **more favorable performance-efficiency frontier**, achieving a higher level of performance for any given computational budget. This analysis compellingly demonstrates our method's superiority, establishing it as a **more advanced and practical choice** for maximizing generation quality.

---

> > > > ### Author Response · Authors · 2025-11-23
> > > > **Response to Reviewer dNUy (4/4)**
> > > >
> > > > ## Q2: Effect on compositional prompts, text rending and texture
> > > > We are pleased to report that S²-Guidance demonstrates significant and measurable improvements in all three areas: compositional reasoning, text rendering, and the generation of high-frequency textures.
> > > >
> > > > (1)  **On Long Compositional Prompts:**
> > > >     S²-Guidance excels at handling complex prompts with multiple objects and relationships, a common failure point for standard guidance.
> > > >
> > > > * **Quantitative Evidence:** This is systematically evaluated on T2I-CompBench (**Table 1**), a benchmark specifically designed for compositional generation. Our method substantially outperforms CFG. For instance, on the SD3 model, S²-Guidance boosts scores for compositional attributes like Color (+6.02), Shape (+7.51), and Texture (+4.32), showcasing superior compositional understanding.
> > > > * **Qualitative Evidence:** **Figure 5** shows clear examples, such as "Cat wearing cowboy hat rides on corgi..." where S²-Guidance correctly composes the scene while other methods fail. Furthermore, **Figure 13 in the Appendix** provides a clear example of attribute binding ("oval sink and rectangular mirror"), where our method correctly assigns attributes while others struggle.
> > > >
> > > > (2)   **On Text Rendering:**
> > > >     Our method preserves and often enhances the text-rendering capabilities of the base model by correcting for guidance-induced artifacts.
> > > >
> > > > * **Qualitative Evidence:** In **Figure 1**, the prompt for the "Floating Castle" image explicitly requests text: "...form the text ’S2 Guidance Is All You Need’...". Our method successfully generates a coherent and stylized phrase that aligns with the prompt. In contrast, the standard CFG output shows illegible artifacts, demonstrating that our guidance helps rather than hinders this difficult task by leveraging the strong text-rendering abilities of the base model (**SD3**).
> > > >
> > > > (3)  **On High-Frequency Textures:**
> > > >     A primary advantage of S²-Guidance is the enhancement of fine details and textures, directly combating the over-smoothing effect sometimes seen with CFG.
> > > >
> > > > * **Quantitative Evidence:** This is validated by the Texture score on T2I-CompBench (**Table 1**), where our method improves the score from 52.45 to 56.77 on **SD3**. Furthermore, our user study (**Appendix B.3, Figure 11**) provides strong perceptual support: when evaluating "Detail Preservation," human evaluators significantly preferred our method (32.5%) over all baselines, including CFG (18.3%).
> > > > * **Qualitative Evidence:** Many figures showcase this. For instance, **Figure 1** highlights "richer artistic detail" and the "astronaut’s transparent helmet." **Figure 5** demonstrates superior rendering of textures like the "leather jacket and metallic zipper."
> > > >
> > > > **In summary,** the evidence shows that S²-Guidance is not a source of destructive interference but a targeted corrective mechanism. The core idea is to steer the sampling trajectory away from the suboptimal predictions of a weaker sub-network, which often manifest as blurry outputs, incorrect compositions, or incoherent text. By treating these as "failure modes" to be avoided, our method regularizes the sampling process, resulting in demonstrably better compositional coherence, text legibility, and texture fidelity.
> > > >
> > > > ---
> > > >
> > > > Please let us know if any concerns remain unaddressed; we are happy to discuss them.

---

> > > > > ### Comment · Reviewer_dNUy · 2025-11-24
> > > > >
> > > > > I would like to thank the authors for their efforts in providing answers to the questions. Your response has clarified my concerns.

---

> > > > > > ### Author Response · Authors · 2025-11-24
> > > > > > **Thanks for recognizing the value of our work!**
> > > > > >
> > > > > > We sincerely thank you for your constructive feedback and for championing our work by raising the score!
> > > > > >
> > > > > > We are particularly grateful that our rebuttal successfully addressed your initial concerns. Your acknowledgement of our method's *theoretical analysis*, its *low-variance stability under randomness*, the *superior performance-efficiency trade-off*, and its *effectiveness in video generation* is immensely encouraging. Your thoughtful engagement has been extremely helpful in strengthening our paper.
> > > > > >
> > > > > > Once again, thank you for your valuable time and support.

---

### Author Response · Authors · 2025-11-26
**General response to reviewers**

We extend our sincere gratitude to all the reviewers (**R1-dNUy**, **R2-amct**, **R3-2Qir**, and **R4-PEh4**) for their insightful and considerate reviews, which have been invaluable in strengthening our manuscript.

We are pleased to acknowledge that the reviewers recognized the **clear and insightful motivation** of our work (**R4**), the **elegant, effective, and practical design** of our method (**R1**, **R3**, **R4**), our **broad experimental coverage** (**R2**, **R4**) and **consistently positive results** (**R2**, **R4**), the method's **high efficiency** (**R1**, **R3**, **R4**), and the **well-written and clear presentation** of our paper (**R1**, **R4**).

In direct response to your thoughtful comments, we have methodically addressed each point in our individual responses. Below is a summary of the major revisions made to the manuscript:

*   We conducted **a quantitative variance analysis** to demonstrate the high stability and negligible variance of our method.
*   We added **a detailed analysis of the computational cost** to offer a transparent view of its practical overhead.
*   We introduced **a performance-vs-efficiency analysis** to demonstrate its superior performance-efficiency trade-off.

We would like to express our sincere gratitude once again for your constructive suggestions, which have been crucial for improving the manuscript.

We have carefully revised the paper to incorporate all your comments. **We are committed to ensuring all your concerns are fully addressed and eagerly look forward to further discussion.** Should any aspects of our work remain unclear, we **welcome any further feedback** to help us improve the manuscript.

---

### Author Response · Authors · 2025-12-01
**Summary to Area Chair**

Dear Area Chair,

*First and foremost, we would like to express our sincere gratitude for the significant time and effort you have dedicated to coordinating the review process, especially given the substantial workload and the recent unexpected circumstances.*

Following our comprehensive rebuttal, we are pleased to report that the process successfully resulted in a `positive recommendation for acceptance from all reviewers` early in the discussion phase (`by November 24, well before the recent unexpected circumstances`).

To assist with your final decision, we provide a summary of the **(1) consensus timeline**, **(2) core contributions**, **(3) reviewer recognition**, and **(4) additional validations and analyses** during the rebuttal below.

## **Part (1/2)**

**(1) Timeline of Consensus:**

* **Strong Initial Basis:** The paper **received strong support from the start**, securing **positive recommendations for acceptance** from **three out of four reviewers** in the initial round (`6,6,6`).
* **Full Consensus Reached (Nov 24):** Reviewer dNUy, **the only reviewer** who initially held reservations, **explicitly acknowledged** that our responses and new results **resolved their concerns**. This constructive discussion resulted in a `score increase (raised to 6)` on November 24, finalizing the `positive consensus (6,6,6,6) from all reviewers` well before the recent unexpected circumstances.

**(2) Summary of Contributions:**

In this work, we identify the suboptimal predictions in Classifier-Free Guidance (CFG) and propose S$^2$-Guidance (Stochastic Self-Guidance), a **training-free** and **plug-and-play** guidance algorithm for diffusion transformers. Our method addresses the challenge of **low fidelity and semantic incoherence** by leveraging **stochastic block-dropping** to construct intrinsic "weak" sub-networks for effective self-correction during the denoising process. This mechanism **effectively guides the model away from potential low-quality predictions, thereby improving fidelity**. Furthermore, our approach **can be seamlessly adapted to various generative models**.

Comprehensive qualitative and quantitative experiments **across multiple models and benchmarks**—spanning **class-conditional** ImageNet generation, **text-to-image**, and **text-to-video** tasks—demonstrate that S$^2$-Guidance delivers **superior performance and efficiency**, consistently surpassing CFG and other advanced guidance strategies.

**(3) Reviewer Recognition:**

We are pleased to acknowledge that the reviewers (**R1-dNUy**, **R2-amct**, **R3-2Qir**, and **R4-PEh4**) recognized the:
* **"Clear and insightful motivation"** (**R4**).
* **"Elegant"** and **"practical design"** of the proposed method (**R1**, **R3**, **R4**).
* **"High efficiency"** of our approach alongside its **"effective"** performance (**R1**, **R3**, **R4**).
* **"Broad experimental coverage"** and **"consistently positive results"** (**R2**, **R4**).
* **"Well-written and clear presentation"** of the paper (**R1**, **R4**).

---

> ### Author Response · Authors · 2025-12-01
> **Summary to Area Chair**
>
> ## **Part (2/2)**
>
> **(4) Additional Validations and Analyses:**
>
> In direct response to the constructive feedback, we have provided **detailed point-by-point responses** to address every concern. Here, we summarize the **major updates**:
>
> * **Stability Verification:**
>
>     To address concerns regarding **sensitivity to randomness and trajectory deviation**, we conducted a **quantitative variance analysis** and provided **additional visualizations**, demonstrating the **high stability and negligible variance** of our method.
>
> * **Theoretical Verification:**
>
>     To address concerns regarding the **theoretical grounding (e.g., unbiasedness) and design rationale**, we provided **additional theoretical derivations**, rigorously verifying the **rationality and reliability** of our proposed method.
>
> * **Temporal Consistency Verification:**
>
>     To address concerns regarding **temporal flickering and identity shifts in video generation**, we reported **comprehensive VBench metrics**, demonstrating that our method **improves Subject Consistency and Temporal Flickering** while significantly boosting **Dynamic Degree**, delivering **superior overall video quality**.
>
> * **Computational Cost Analysis:**
>
>     To address concerns regarding **specific resource requirements (e.g., FLOPs, memory)**, we added a **detailed analysis of the computational cost**, offering a **transparent view of its practical overhead**.
>
> * **Performance-Efficiency Trade-off:**
>
>     To address concerns regarding the **computational overhead of the extra forward pass**, we introduced a **performance-vs-efficiency analysis**, demonstrating that our method achieves a **superior trade-off** (higher performance for any given computational budget).
>
> `We believe we have fully addressed the concerns raised by the reviewers through these analyses.` We have already incorporated these improvements into the updated version of the paper.
>
> *Once again, thank you for your hard work and service to the community, especially during this challenging period. We genuinely appreciate your dedication to maintaining the integrity of the review process despite the recent difficulties.*
>
> Best regards,
>
> The Authors

---

### Meta-Review · Area_Chair_i3Yb · 2026-01-09

**Summary:**

This paper introduces a simple, training-free stochastic guidance method for diffusion models by activating internal “weak” predictors through stochastic block dropping during inference. The approach is elegant and easy to implement, requiring no auxiliary models or additional training, and can be seamlessly integrated into existing classifier-free guidance (CFG) pipelines. The idea is intuitive yet practically appealing, and the paper is clearly written and easy to follow.

A notable strength of this work is its minimalist design with consistent empirical benefits. The proposed method requires only one additional stochastic forward pass per denoising step, which is considerably lighter than many alternative guidance or self-ensemble strategies. Extensive experiments across multiple transformer-based diffusion backbones for both image and video generation demonstrate consistent improvements over standard CFG, supported by clear qualitative results and ablation studies on key hyperparameters such as the drop ratio and guidance weight. The toy Gaussian mixture experiments provide a helpful diagnostic perspective on CFG’s limitations and effectively motivate the proposed self-correcting mechanism.

That said, the contribution is incremental rather than paradigm-shifting, and several aspects would benefit from further clarification and strengthening. The theoretical framing, particularly the Bayesian interpretation provided in the appendix, remains largely interpretive. Key assumptions—such as the unbiasedness of the stochastic guidance estimate or the small variance of a single stochastic sub-network—are not formally justified, and the connection to improved score estimation or likelihood remains intuitive rather than rigorous. While a full theoretical treatment may be beyond the paper’s scope, a clearer discussion of variance, stability, or concentration behavior would strengthen the contribution.

There are also open questions regarding robustness and efficiency, especially for video generation. The stochastic block dropping introduces randomness that may affect temporal consistency, and some results suggest slight degradation in subject or background consistency. Explicit analysis of output variance, temporal stability metrics, and clearer reporting of FLOPs, memory usage, or wall-clock time relative to standard CFG would help better substantiate the efficiency claims.

Finally, while the paper positions itself as a general solution for weak-model guidance, the evaluation is limited to transformer-based diffusion architectures. Discussion of applicability to UNet/CNN backbones, or clarification of architectural constraints, would improve the generality claims. In addition, the paper would benefit from stronger contextualization with closely related prior work, particularly Score-based Adaptive Guidance (SAG, ICCV 2023) and Perturb-and-Guide (PAG, ECCV 2024). These methods share conceptual similarities in using degraded or perturbed predictions for guidance, and should be explicitly discussed and cited in the final version, ideally with direct empirical comparisons where feasible.

In summary, this paper presents a practically useful, easy-to-adopt improvement to CFG that delivers consistent empirical gains with minimal overhead. While the novelty is modest and several analyses are missing, the simplicity, effectiveness, and broad experimental validation make it a valuable contribution. With clearer positioning relative to prior work (including SAG and PAG) and additional discussion of variance, efficiency, and temporal stability, the work would be well-suited for acceptance.

**Reviewer Concerns:**

Reviewer dNUy's concerns were addressed.

**Reviewer Scores:**

Reviewer dNUy might change the score.

---

### Decision · Program_Chairs · 2026-01-26

Accept (Poster)